# The Panasqueira Rare Metal Granite Suites and Their Involvement in the Genesis of the World-Class Panasqueira W–Sn–Cu Vein Deposit: A Petrographic, Mineralogical, and Geochemical Study

**Christian Marignac** [1] , **Michel Cuney** [1,*] , **Michel Cathelineau** [1] , **Andreï Lecomte** [1] , **Eleonora Carocci** [1] and **Filipe Pinto** [2]

1   Université de Lorraine, CNRS, CREGU, GeoRessources, F-54000 Nancy, France; christian.marignac@univ-lorraine.fr (C.M.); michel.cathelineau@univ-lorraine.fr (M.C.); andrei.lecomte@univ-lorraine.fr (A.L.); eleonora.carocci@univ-lorraine.fr (E.C.)
2   Panasqueira Mine, Beralt Tin and Wolfram, Barroca Grande, 6225-051 Aldeia de São Francisco de Assis, Portugal; filipe.Pinto@beraltportugal.pt
*   Correspondence: michel.cuney@univ-lorraine.fr

**Abstract:** Elucidation of time-space relationships between a given wolframite deposit and the associated granites, the nature of the latter, and their alterations, is a prerequisite to establishing a genetic model. In the case of the world-class Panasqueira deposit, the problem is complicated because the associated granites are concealed and until now poorly known. The study of samples from a recent drill hole and a new gallery allowed a new approach of the Panasqueira granite system. Detailed petrographic, mineralogical, and geochemical studies were conducted, involving bulk major and trace analyses, BSE and CL imaging, EPMA, and SEM-EDS analyses of minerals. The apical part of the Pansqueira pluton consisted of a layered sequence of separate granite pulses, strongly affected by polyphase alteration. The use of pertinent geochemical diagrams (major and trace elements) facilitated the discrimination of magmatic and alteration trends. The studied samples were representative of a magmatic suite of the high-phosphorus peraluminous rare-metal granite type. The less fractionated members were porphyritic protolithionite granites (G1), the more evolved member was an albite-Li-muscovite rare metal granite (G4). Granites showed three types of alteration processes. Early muscovitisation (Ms0) affected the protolithionite in G1. Intense silicification affected the upper G4 cupola. Late muscovitisation (Fe–Li–Ms1) was pervasive in all facies, more intense in the G4 cupola, where quartz replacement yielded quartz-muscovite (pseudo-greisen) and muscovite only (episyenite) rocks. These alterations were prone to yield rare metals to the coeval quartz-wolframite veins.

**Keywords:** tungsten-tin deposit; Panasqueira; granite; mineralogy; geochemistry; pseudo-greisen

## 1. Introduction

The genesis of hydrothermal W and Sn deposits is generally related to the emplacement of so-called specialized granites, enriched in a series of incompatible elements, such as Rb, Cs, Li, Sn, W, Ta, Nb (e.g., [1–3]). The tin-tungsten veins occur in the exo- and endo-contact zone of the granites. However, the genetic link for the source of the metals between these granites and the deposits is debated (e.g., [4–10]). The Panasqueira W–Sn(Cu) deposit in Central Portugal, with ab initio endowment of ≥115 kt W, is one of the largest W deposits in western Europe. W-ore occurs as a set of subhorizontal quartz-wolframite veins in the thermal metamorphism envelope related to a concealed

granite body [11]. Since the 1970s, the Panasqueira granite has been considered the main control for the formation of the Panasqueira W–Sn(Cu) mineralization. Most of the available works [11–16] have mostly addressed major element geochemistry, mainly based on drill-hole data, in general, with a limited number of samples, not located for most of them, commonly affected by deep hydrothermal alterations (greisenisation), and lacking a detailed petrographic-mineralogical description. For these reasons, it has been until now challenging to assign the Panasqueira granite to any reference granite trend. Because of the scarcity in trace element data, it has not been easy to understand the granite/vein system relationships. Fortunately, the Panasqueira mine company has recently performed a detailed analysis of a 200 m long drill section in the granite body [16]. Besides, it has authorized several teams, including ours, to select samples of the freshest granites along this section. This sampling gave us the opportunity of a detailed petrographic-mineralogical analysis [17] and to get additional geochemical data. Besides, the apical part of the pluton could be sampled and studied, thanks to a new mine gallery dug at a shallow level [18]. The most reliable geochemical data from literature were finally integrated into the new set of data. Finally, the overall consideration of Panasqueira granite suites, especially their magmatic evolution and related fluid-rock interactions, allowed us to examine the possible genetic links between magmatic and hydrothermal events at Panasqueira.

## 2. Regional Geology

The Panasqueira W–Sn–(Cu) ore deposit is located in the province of Beira Baixa (Central Portugal). It occurs in the Central Iberian Zone (CIZ), which represents the axial zone of the Iberian Variscan Belt (Figure 1A). The CIZ is mainly composed of a thick sequence of late Ediacaran-Cambrian schists and graywackes, the Schist-Greywacke-Complex or Beira group, uncomfortably overlain by the Early Ordovician Armorican Quartzite [19]. The tectonic and thermal evolution in a collisional context encompassed the whole Carboniferous, ending in the Early Permian [20–22]. Following initial crustal thickening, the late Carboniferous evolution was characterized by large-scale crustal melting, resulting from the combination of mid-crustal post-thickening heating and mantle-derived heat input at the base of the continental crust [23]. As a consequence, syn-to post-kinematic granite plutons were emplaced in the 315–295 Ma interval [24–27]. Simultaneously, a protracted sequence of episodic intra-crustal transcurrent deformation was initiated at ca. 315–310 Ma by the development of km- to 10 km-sized NW-SE to N-S upright F3 folds, with an S3 axial-plane schistosity, followed by left-lateral (ca. 309–305 Ma), then dextral (ca. 304–295 Ma) faulting [28], and references therein. Numerous mineralized systems were active in the CIZ in close association with late Carboniferous tectonics and granitoid emplacement. More than 150 deposits are known, corresponding to both peri- and intra-granitic rare metal (Nb-Ta, Li, W, Sn) and shear-zone hosted (Au–Ag–As–Sb–Pb–Zn–Cu) systems [25,29–31].

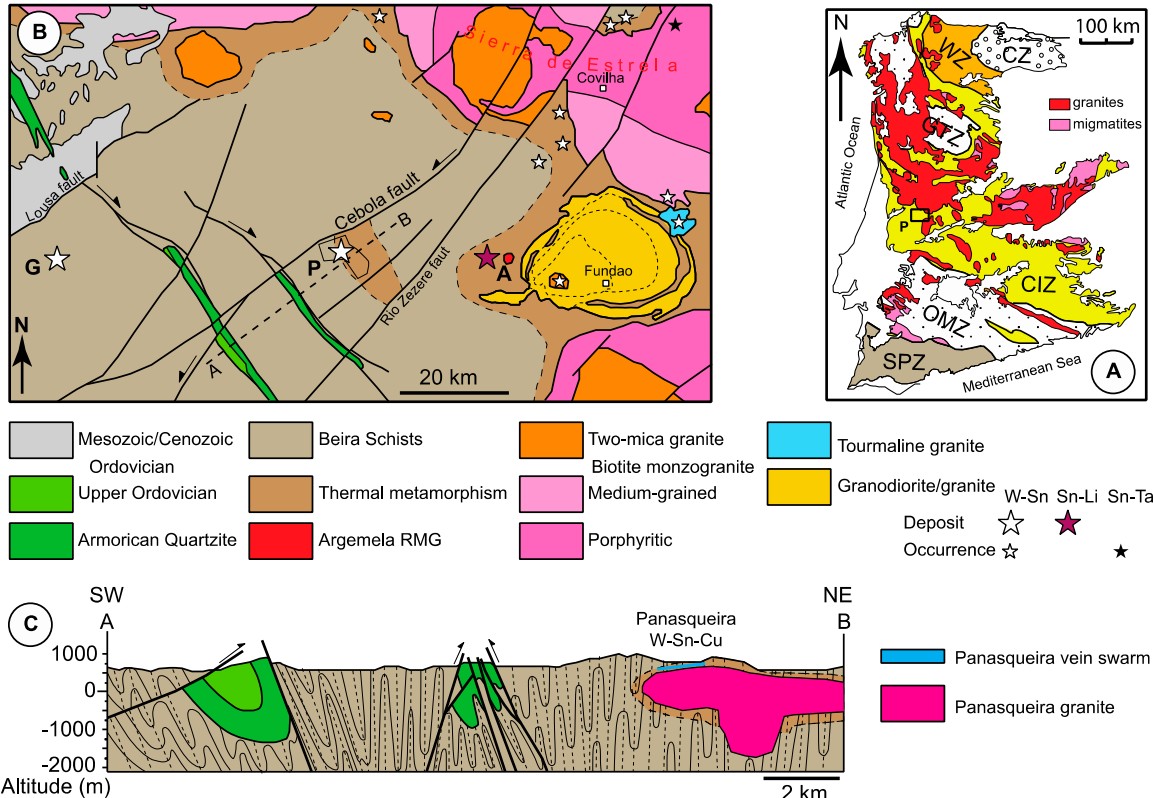

**Figure 1.** Geological setting of the Panasqueira deposit. (**A**) The regional setting, with the main lithotectonic units (simplified from [20]): SPZ (South Portuguese Zone), GTZ (Galicia-Tras-os-Montes Zone), OMZ (Ossa Morena Zone), CZ (Cantabrian Zone) allochthonous units; CIZ (Central Iberian Zone), WZ (Western Asturian-Leon Zone) autochthonous units. P: location of Figure B. (**B**) Geologic map of the Panasqueira area (located in Figure A) adapted from [12–15]. A: Argemela; G: Gois; P: Panasqueira. Line A–B: section of Figure C. (**C**) A-B geologic cross-section, simplified from [15].

## 3. Panasqueira W–Sn–(Cu) Deposit

Panasqueira deposit is hosted in the core of a late Carboniferous anticlinorium of Beira Schist, bordered by narrow synclines of Ordovician Armorican Quartzite, to the south-west of the late- to post-tectonic Serra da Estrela granitic complex (Figure 1B,C). Around 10 km to the east of Panasqueira, the Argemela cupola outcrops. It is a small rare metal granite (RMG) body associated with tin mineralization and minor wolframite veins [32,33]. The Beira schists have been metamorphosed to the lower greenschist facies. Early irregular milky quartz veins called "Seixo Bravo", sub-concordant with the vertical foliation of the micaschists, are late kinematic. They are especially abundant in the Panasqueira mine area. They are always barren.

At Panasqueira, the presence of a thermal metamorphic aureole represented by biotite-cordierite spotted schists with andalusite around the granitic contact indicates the presence of a granitic intrusion at depth (Figures 1B and 2A) [34]. These mineral assemblages constrain the pressure to have been no more than 300 MPa at the time of emplacement of the Panasqueira Granite [35–37]. The upper part of this granite, variably affected by greisenisation (greisenised cupola), is exposed in a few underground mine galleries and has been intersected by several drill-holes, of which only one (SCB2) has been preserved (Figure 2C).

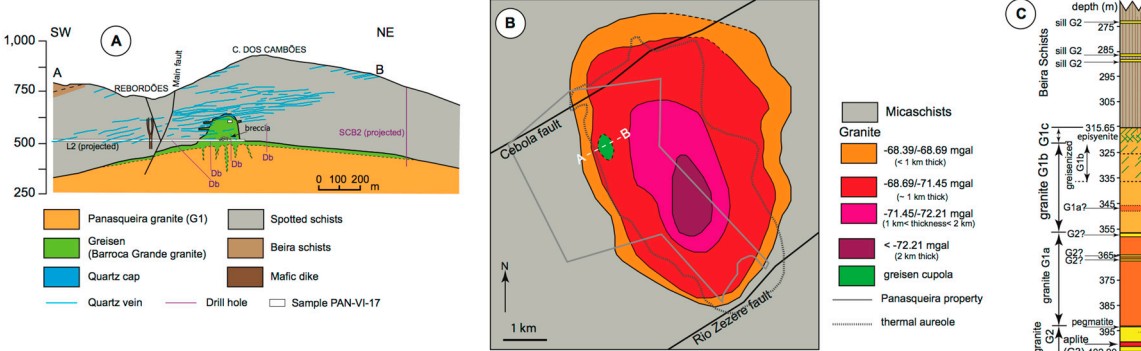

**Figure 2.** Panasqueira granite architecture. (**A**) The accepted petrography of the upper granite body, according to [11,12]. Section A–B is located in Figure B. Location of the old drill holes, as well as the recent SCB2 hole and new L2 gallery, are indicated. The vertical greisen roots are drawn as proposed by [12], but without any observational base (no intersection by the figured drill holes). (**B**) Gravimetric data interpretation of the shape of the granite body, according to [34]. (**C**) Synthetic petrographic profile of the SCB2 drill-hole.

Sets of shallow-dipping quartz veins occur within the envelope of thermal metamorphism and host the economic W–Sn(Cu) mineralization. They are concentrated above the cupola and overprint it. They crosscut the Seixo Bravo veins. The vein system extends over an area of about 10 km$^2$, for a vertical extension of about 300 m [7–11]. A late ENE-WSW and NNW-SSE sub-vertical fault system cross-cuts the granite intrusion and the veins [37]. This deposit has been the subject of many studies, revealing a complex history [7,11,12,15,37–45]. The rutile associated with a strong tourmalinization of the wall-rocks preceding the main wolframite deposition stage [46] has been dated at 305.2 ± 5.7 Ma using the U–Pb technique by LA-ICP-MS [44].

## 4. Granite Architecture: Previous Works and New Findings

At the Panasqueira tungsten deposit, a concealed granite is only revealed at the surface by a conspicuous aureole of contact metamorphism and a series of NW-SE trending aplitic dikes, either concentrated in the Cabeço do Piao area where they are overprinted by W-bearing quartz veins, or sporadically occurring along the eastern border of the aureole [11]. Early mine works have revealed the Panasqueira Granite as a greisenised cupola [33,47–49]. Subsequently, several drill holes reached larger bodies made of two-mica porphyritic granite, so-called G1 (Figure 2A) [12]. According to recent geophysical modeling [34], the main body corresponds to a relatively small and thin granitic laccolith only about 1 km thick, 6.5 km long, and 4.5 km large with a conspicuous keel rooted down to 2 km and elongated in the NW-SE direction. A rather flat top on which a cupola, also elongated in the NE-SW direction, was offset from the root (Figure 2B). The volume of the intrusion might be estimated at 32 km$^3$.

Exploitation galleries reached the cupola at several levels (L1, L2, and L530), in particular, a new gallery at the L2 level, which allowed new observations. A small "quartz cap" was encountered at the very top of the cupola (Figure 3A) in a gallery now without access, and is only known by a short description given by [11]. Some drill-hole sections close to the cupola encountered granite layers separated by 10–15 m thick micaschist screens (Figure 3B), layers interpreted as sills protruding from the cupola [49,50]. Such sills also occur at the L1 level, where they crosscut the Seixo Bravo veins [15]. Thin sills of a strongly altered granite were observed in the schists within the first tens of meters above the granite roof in the SCB2 drill hole (Figure 2C).

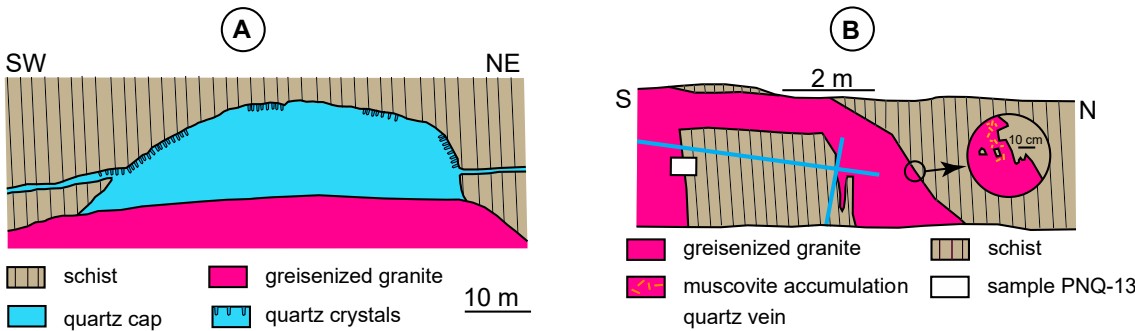

**Figure 3.** Details of the greisen cupola. (**A**) Quartz cap at the top of the greisen cupola (redrawn and interpreted from [11]). The big quartz crystals are inferred from the original description of [11], having not been drawn by them. (**B**) Sketch of the (greisenised) G4 granite intrusive contact at the L2 level (observations at the L2 level-2018).

A section of the shallowest parts of the Panasqueira granite, until now considered as the reference, has been reconstructed by [12], using a series of drill holes that were not well preserved (Figure 2A). According to [12], following [11], a thin topmost layer of the Panasqueira granite is a greisen, considered to continuously pass to the greisen cupola, both resulting from the alteration of the main porphyritic granite. Almost all of the followers, including [15], have adopted this view. It is worth noticing that the vertical greisen funnels drawn by [12] through the main granite below the greisen layer are highly speculative, having not been crossed by the drill holes, as represented in the Bussink [12] section. Nonetheless, in 2006, a study [14] revisited the cupola at the L2(P4) level and identified a Barroca Grande granite, differing from the Panasqueira biotite granite and more akin to the Argemela rare metal granite, and having suffered greisenisation. The 1979 study [11] brought to attention the presence at the bottom of the greisenised cupola (L2 level) of a matrix-supported breccia, with Beira schist clasts (up to 2 m and more) and a greisen matrix. An equivalent of this breccia, but with only quartz vein clasts, was encountered at the same level in the new gallery at the L2 level (Figure 2A). Otherwise, small angular hornfels enclaves are said to have been common, close to the granite contact with the surrounding schists [11]. A drill hole described by [51] has encountered a schist panel 50 m below the granite top.

Only short accounts of the granite petrography from the old drill holes can be found [11,12]. Some of the old drill cores are now in bad condition. The SCB2 hole was sampled by [15] and then by our team. From the upper contact with the Beira schists, the drill hole crossed three granite units (Figure 2C), from top to bottom, a G1 suite with three facies: a more or less greisenised "equigranular" granite (G1b unit), with a more evolved topping facies (G1c), and a porphyritic two-mica granite (G1a unit); a leucogranite (G2 unit); a fine-grained leucogranite ("aplite", G3 unit). Rather clear-cut magmatic contacts separated all facies. Thin leucogranite layers, interpreted as parts of G2, were observed in the G1a unit and at the G1a to G1b transition. A possible recurrence of the G1a facies was present in the G1b unit. A thin quartz-feldspar pegmatite layer was intercalated at the G2 to G1a boundary.

## 5. Materials and Methods

Five samples selected along with the SCB2 drill hole representative of the less altered granites (see Figure 2C) [17] and five from the new L2 level gallery (four greisens and one muscovite) [18] were analyzed for this study. These analyses were to be compared with the samples of [15,16], also coming from the SCB2 drill-hole. The samples of [15] were not located, but came from the SCB2 drill hole, encompassing "unaltered" and "greisenised" G1 granite, and the apical greisen, and from the L1 level in the greisenised cupola (called "greisens" in Launay's work [15]). File S2 describes the methods.

## 6. Results: Petrography

*6.1. The Greisenised Cupola*

6.1.1. Observations at the L2 Level

At the L2 level, the intrusive contacts of the greisenised granite (G4, defined in the geochemistry) with the surrounding metamorphic rocks showed a combination of steep (~N150°E) and flat interfaces, both oblique onto the near-vertical schistosity of the wall rock at ~N140°E (Figure 3B). A series of quartz enclaves, constitutive of a granite matrix-supported breccia, was observed, as well as a variety of veins, some of them cross-cutting the granite edge.

At the contact, in the metamorphic rocks, a new generation of a zoned protolithionite developed pervasively along the schistosity (Figure 4A). Between the granite and the schist, a thin millimetric rim of a very fine-grained granite (aplite) formed (Figure 4B). The aplite consisted of quartz-albite-muscovite-protolithionite and seemed overprinted by the protolithionite developed in the wall rock. A few small ilmenites and rutile crystals, the latter with a very low content in rare metals (up to 0.37 wt% Nb, traces of Ta), were disseminated in the aplite. The contact granite itself was greisenised as the rest of the granite body, with, however, a lesser quantity of relict albite.

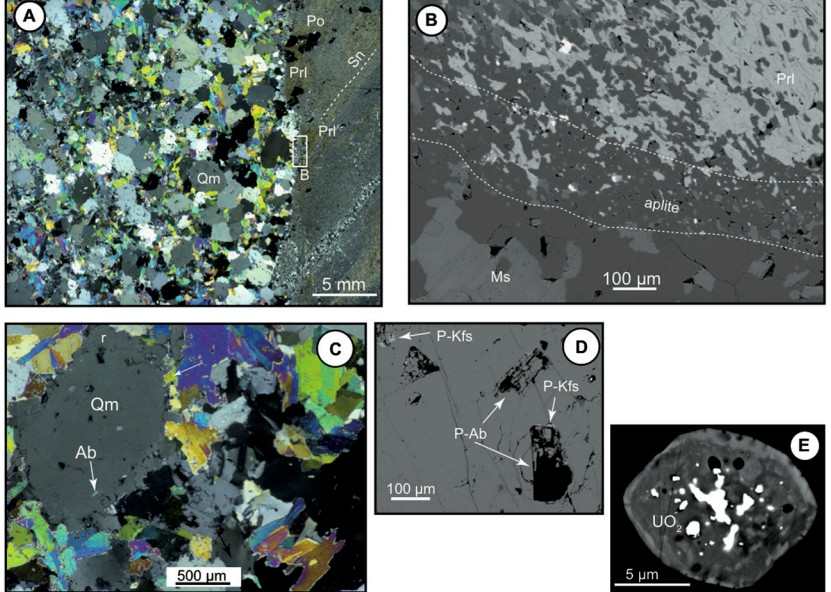

**Figure 4.** Photomicrographs and SEM-BSE images of the greisen cupola at the L2 level. (**A**) Granite (greisenised)/wallrock contact (cross-polarized light, sample PNQ-13a). Starting from the contact, protolithionite (Prl) is invading the wall rock, thanks to the schistosity (Sn); Po: pyrrhotite. (**B**) SEM-BSE image of the granite-schist contact (located in A). A thin aplite rim is intercalated at the contact. Together with quartz and albite, it contains minute inclusions of muscovite, protolithionite, and rutile (bright spots). (**C**) Relic embryonic snowball quartz in the greisenised G4 granite (sample PNQ-13a, cross-polarized light). The small included albite (Ab) in the magmatic quartz (Qm) defines the "snowball" texture. Note the secondary quartz expansions (r) around the magmatic quartz, and the interplay of muscovite and this rim (arrow). (**D**) SEM-BSE image of included P-enriched albite (P-Ab) and Kfs (P-Kfs) in the magmatic quartz (sample PNQ-19a). (**E**) Zoned zircon in the muscovite at the L2 level (BSE, sample PNQ-17a). Note the numerous uraninite inclusions. The rim is depleted in Hf and U-rich. Note the great difference with the G1 zircon (Figure 10D).

The greisenised granite consisted mainly of an assemblage of quartz and muscovite (Figure 4A), where only relics of the pristine granite were preserved. Among these were rather common rounded magmatic quartz, displaying an aureole of festooned digitations and internal zoning characteristic of

an embryonic snowball structure, marked by concentric feldspar microliths (Figure 4C). The microliths were mainly albite, with minor K-feldspar, the latter either capping albite or being isolated (tending to be more external relative to the albite) (Figure 4D). Other relicts consisted of lace-shaped ghosts of albite (0.2–0.4 mm) invaded by quartz (Figure 5). Albite relicts were pure, with only occasional minor Ca and K contents. They tended to be more tenuous, close to the contact. Skeletal relicts of euhedral topaz crystals were not uncommon (Figure 5B), testifying for the presence of topaz in the pristine mineral assemblage. Muscovite was typically zoned (Figure 6A,C). Accessory minerals were mainly apatite and zircon.

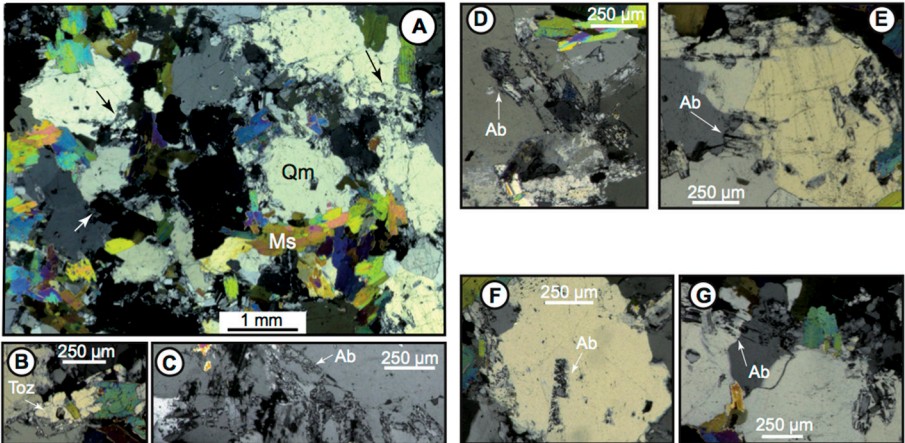

**Figure 5.** Photomicrographs and SEM-BSE images of albite habit in the greisenised G4 granite (L2 level) (cross-polarized light, sample PNQ-19a). (**A**) Overall view of the greisen. The rock mainly consists of quartz and muscovite (Ms), with albite ghosts overprinted by quartz (arrows). Qm relic snowball quartz. (**B**) Skeletal topaz included in quartz. (**C**–**G**) Lace-shaped ghosts of albite crystals invaded by quartz. Ab: albite, Toz: topaz.

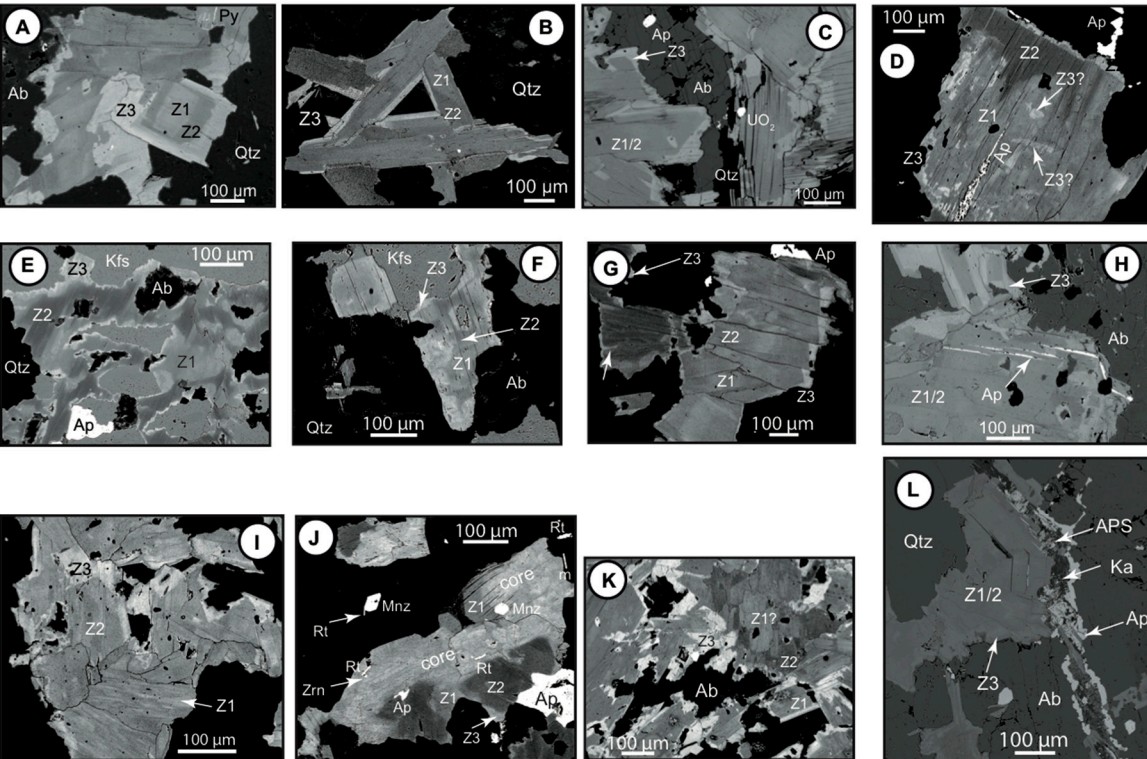

**Figure 6.** SEM-BSE images of muscovite zoning. The zoning pattern consists of three zones with a grey core (Z1), a darker first rim (Z2), and a bright second rim (Z3). Ab albite, Ap apatite, APS alumina-phosphate-sulfate, Kfs K-feldspar, Rt rutile, Mnz monazite, Qtz quartz, Zrn zircon. **A** to **C**: muscovite from the greisenised G4 granite. (**A**) L2 level (sample PNQ-19a). A clear succession of the three zones (Z1 to Z3) is visible, although patchy zoning features are also evident. (**B**) L0 level (sample PAN-VI-17a2-1). The Z1 to Z3 succession is well seen. (**C**) L2 level, contact with the wall rock (sample PNQ-13a). Privileged Z3 development occurs along the muscovite/albite boundary. Note the uraninite inclusion. (**D,E**) Muscovite in granites from the G1 suite. (**D**) Muscovite strips in the Kfs from the porphyritic granite G1a (sample PAN16-6). Patchy zoning with clear superimposition of Z2 onto Z1. (**E**) Greisenised G1b granite (sample PAN16-4a). Note the ingress of Z3 onto the patchy Z1/2 assemblage. (**F**) Altered G2 granite (sample PAN16-8). Note the concentric Z1 to Z3 pattern. Arrow: Z3 invasive along cleavage. (**G**) Altered leucomicrogranite (aplite) G3 (sample PAN16-9). Note the conspicuous Z3 rim and the residual character of the Z1 zone within the main Z2 zone. (**H,I**) Possible relics of an altered magmatic Li-mica. (**H**) In some crystals from the greisenised G1b granite (sample PAN16-3b), the existence of a bright "core", with rutile inclusions, may be tentatively interpreted as resulting from the replacement of a pre-existing Ms0-rutile assemblage developed at the expense of a pristine protolithionite; m is molybdenite. (**I**) Some bright areas in the core of muscovite from the greisenised G4 granite (sample PNQ-13a) may be tentatively interpreted as relics from a pre-existing Li-mica (Z1?). (**J–L**) Muscovite-apatite relationships. (**J**) Altered G2 granite (sample PAN16-8). Note the invasive character of the Z3 zone and the presence of late apatite along the cleavage. Small Fe-rich patches inside Z1 (Z3?) are possible ghosts of an earlier Li-mica. (**K**) Patchy zoning in the muscovite from the greisenised G4 granite near the wall-rock boundary (sample PNQ-13a). The Z3 zone is encroaching the earlier muscovite. Note the clear overprinting by apatite (Ap). (**L**) Microcracks filled with apatite are encroaching the zoned muscovite and the albite in the altered G2 granite. Apatite is replaced by an APS-kaolinite (Ka) assemblage (see Figure 7).

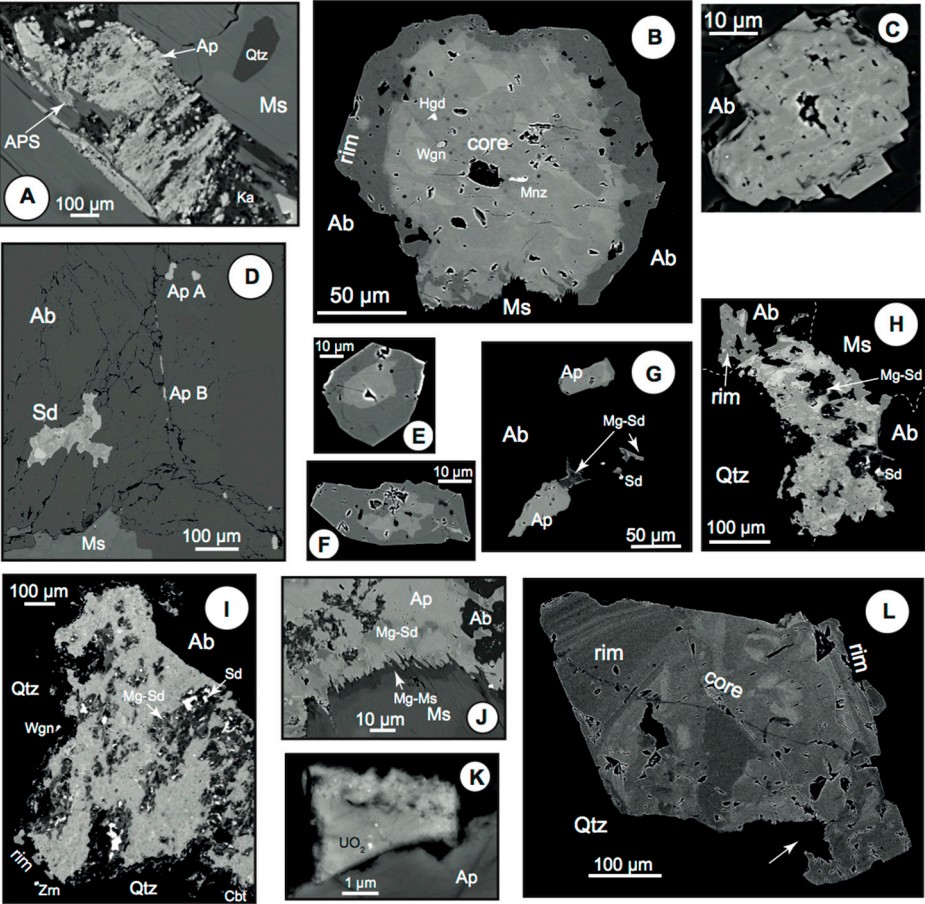

**Figure 7.** SEM-BSE images of apatite in the G2, G3, and G4 granites (BSE images). Ab albite, Ap apatite, APS alumina-phosphate-sulfate, Cbt columbite-tantalite, Ms muscovite, Hgd hagendorfite, Ka kaolinite, Mnz monazite, Qtz quartz, Sd siderite, Wgn wagnerite, Zrn zircon. (**A**) In G2, a transformation of microfissural apatite into an APS-kaolinite assemblage (detail of E in Figure 6). (**B**) Zoned apatite in G2 granite (sample PAN-16-8), with a patchy zoned core and a rim. Note the various phosphate micro-inclusions in the core, and the ragged contact with muscovite. (**C**) Small euhedral Sr-APS crystal included in albite (G2 granite, sample PAN-16-8). The bright internal zones and overgrowth correspond to an increase in the S content. (**D**) Siderite-apatite association in albite from the G3 leuco-microgranite (sample PAN-16-9). Two types of apatite are present: small zoned euhedral crystals (Ap A), like in **E** or **F**, and small homogenous elongated prisms, with external rims in Ap A, apparently following a microcrack in the hosting albite (Ap B). Siderite is zoned, according to variable Mg contents, with the brightest areas being pure Fe-end member. (**E**,**F**) Examples of zoned apatite included in albite (G2 granite, sample PAN-16-8). (**G**) Another example of an apatite-siderite association in albite (G2 granite, sample PAN-16-8). (**H**,**I**) Two zoned apatite crystals in the leuco-microgranite G3 (sample PAN-16-9), with a patchy core and a discontinuous homogenous rim. Note the strong corrosion by internal siderite (Mg-Sd and pure siderite), and the contrast in the boundary habit depending on the host, the boundary being ragged at the quartz contact, with a halo of minute apatite specks, and quite regular at the albite contact. (**J**) In G3, a denticulate contact between apatite and the Mg-rich fringe of a muscovite crystal (sample PAN-16-9). (**K**) In the greisenised G4 granite, overgrowth on apatite of an APS (goyazite-florencite solid solution), with uraninite nano-inclusions (sample PNQ-19a). (**L**) A large crystal of apatite in the greisenised G4 granite (sample PNQ-19a), with a patchy zoned core and a rim displaying a conspicuous OZ. Note the locally contorted boundary (arrow), similar to some apatite habit in greisenised granite G1b (Figure 14). The small zircon crystals (1–5 μm, up to 10–20 μm), included in muscovite or in quartz, are homogenous, or well zoned, with a Hf-rich core (up to 5.3 wt% Hf, Hf# 0.06) and a rim with a lower Hf content (3.2 wt% Hf, Hf# 0.04), but richer in U (up to 2.6 wt% U in the rim, versus 0.6 wt% U in the core).

Apatite formed small homogenous crystals (10–20 μm) included in albite, but larger crystals (up to 400 μm), enclosing albite, and patchy zoned, were also observed, although less common. One of these large crystals displayed a patchy core and an oscillatory zoned rim (see Figure 7L) and might be compared to similar crystals in the G1 suite (see below). The rare occurrence of small overgrowths of an alumina-phosphate-sulfate (APS) with uraninite nano-inclusions (see Figure 7K) was additional evidence for this comparison.

A series of sulfides (arsenopyrite, pyrite, sphalerite, chalcopyrite, galena) and carbonates (siderite) was the result of late growth, likely in correlation with the nearby hydrothermal vein system. Rare pyrrhotite crystals occurred close to the contact with the enclosing rocks. Shielded by pyrite and siderite, they might represent xenocrysts pulled off the surrounding schists, which were pyrrhotite-rich.

Metric-scale isometric patches of a muscovite-rich rock, devoid of quartz, but still containing albite, were sporadically observed within the greisenised mass. This albite was unaffected by dissolution, suggesting possible recrystallization (albite II of [37]) different from the processes found in the greisen. The muscovite appeared either compact or vacuolar. The vugs might contain cassiterite.

The elements of the matrix-supported breccia consisted of either quartz or quartz vein clasts, with a straight upper boundary, presumably pulled off from a former wall rock-vein contact, and a denticulate lower boundary, suggestive of a later overgrowth of quartz when incorporated in the granite melt (Figure 8A,B). The quartz of the vein clasts exhibited evidence of significant plastic deformation, distinctly contrasting with the quartz of the enclosing greisen (Figure 7C), which was suggestive of a "seixo bravo" origin for the vein clasts. As shown in Figure 7C,D, a muscovite fringe obliterated the deformed quartz/greisen boundary, which was then invaded by muscovite aggregates, either along quartz grain boundaries or patches connected to muscovite-sealed microcracks. The muscovite was zoned, in the same way as the muscovite in the greisen (Figure 8F). Dispersed in the quartz, minute zinnwaldite crystals that commonly exhibited evidence for a replacement by muscovite were systematically observed at a short distance (100–200 μm) from muscovite aggregates (Figure 8F,G).

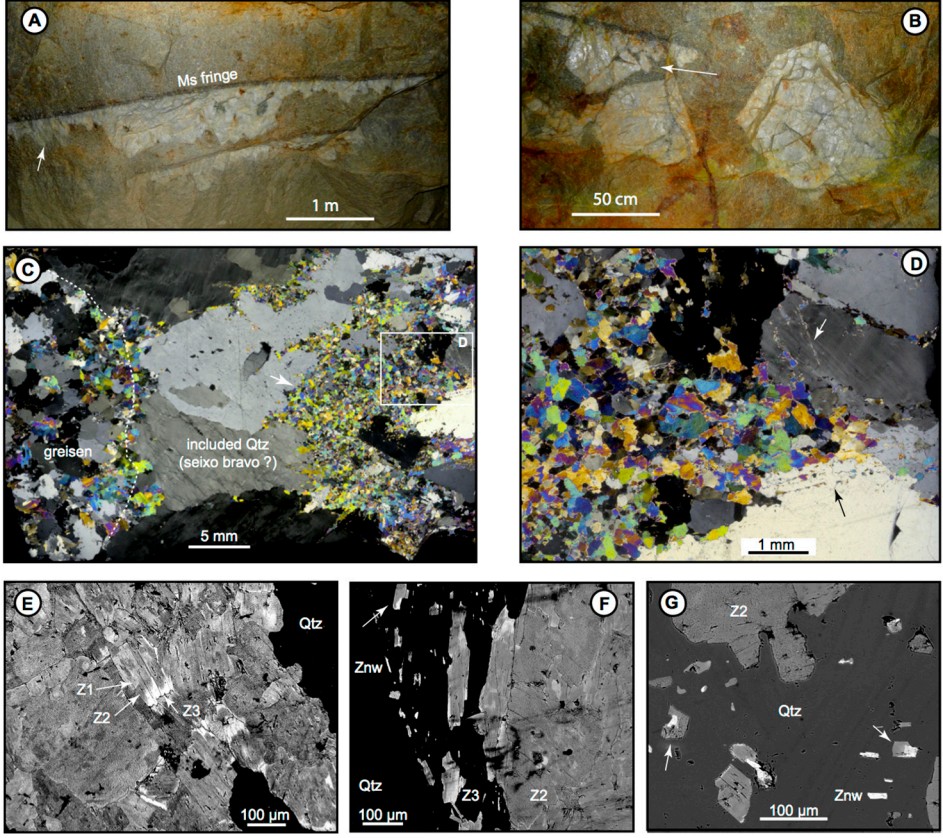

**Figure 8.** Quartz vein enclaves in the greisenised G4 granite from the greisen cupola (L2 level). (**A**) Photograph of dilacerated quartz vein enclaves; note the muscovite fringe and the euhedral quartz growing in the "greisen" matrix (arrow) (underground work, L2). (**B**) Idem; note the infiltration of the magmatic matrix into the vein (arrow). (**C**) Photomicrograph of a detail of the "greisen" contact to the included quartz (cross-polarized light, sample PNQ-21a). Insert shows the location of (**D**). Note the plastic deformation in the included quartz (seixo bravo?); the superimposition of the "greisen" to quartz enclave boundary; and the development of muscovite ("episyenite") within the included quartz. (**D**) Detail of (**C**). Note the trails of tiny muscovite flakes into the deformed quartz (arrows). (**E**) to (**G**) SEM-BSE images of the muscovite-quartz contact around the muscovite in (**D**). (**E**) Zoning in the muscovite: a succession of a clear core (Z1), a darker rim (Z2), making the bulk of the aggregate, and a sporadic bright fringe (Z3). (**F**) Start of a muscovite trail in the quartz. Note the presence of tiny zinnwaldite crystals (Znw), accompanying the small muscovite flakes, and that the latter is growing at the expense of Znw (arrow). (**G**) Halo of tiny Znw crystals (replaced by muscovite: arrows) around the main muscovite aggregate.

### 6.1.2. Observations at the L1 Level

At the L1 level, the greisen was devoid of any feldspar relics, and only rare magmatic quartz relics from the magmatic stage might be found (Figure 9H). This quartz displayed a distinct (although embryonic) snowball texture. The core contained a few albite microliths; in some cases, intergrown with Kfs, whereas, in the rim of Kfs, microliths might occur. One occurrence of a zoned Fe/Mg–Li-muscovite was found as aggregates associated with albite and Kfs. Rare microliths of phosphate of the alluaudite group were also encountered (see below, mineralogy). Muscovite crystals were zoned (Figure 6B). Accessory minerals were mainly apatite (as at the L2 level), with a few zircon crystals. The latter were enriched in hafnium (3.8 to 4.4 wt% Hf, Hf# 0.05 to 0.04), like their counterpart at the L2 level.

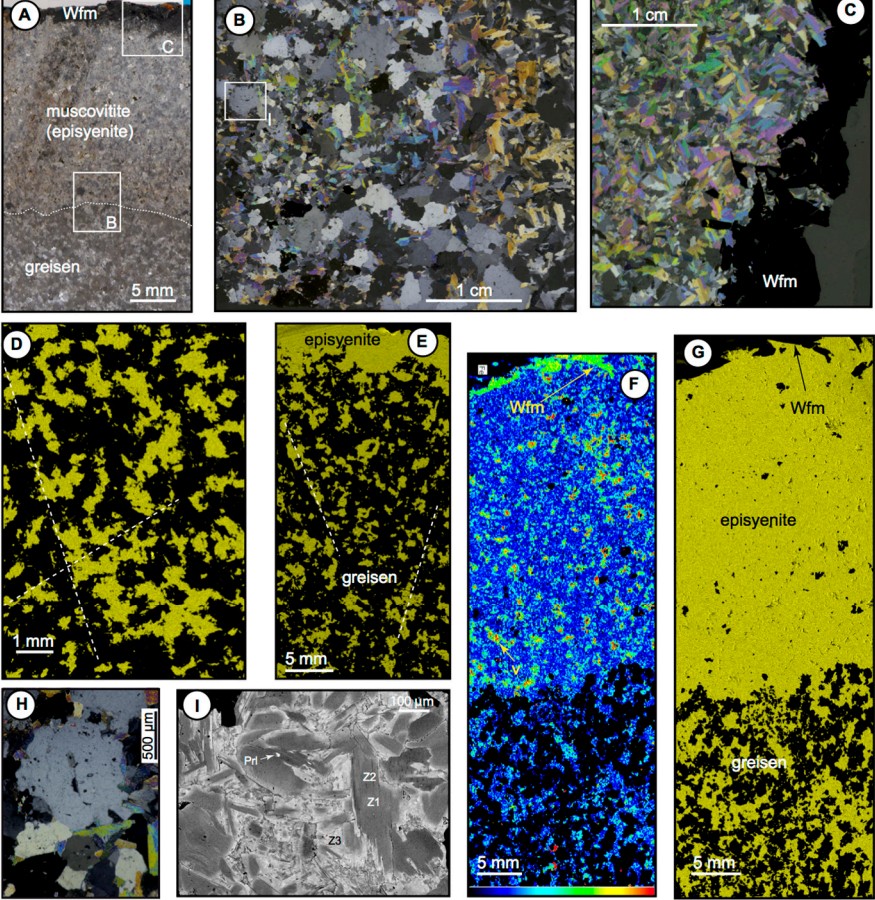

**Figure 9.** Muscovite and muscovite ("episyenite") in the greisen cupola, at the L0 (**A–C**, **F–H**, sample PAN-VI-17), and the L2 (**D,E,I**) levels. (**A**) Overview photograph of the sample, showing the joining of a muscovite-only rock on to a greisen. Note the wolframite (Wfm) fringe in the muscovite. (**B**) Overview photomicrograph of the greisen/muscovite contact (cross-polarized light). (**C**) Overview photomicrographs of the muscovite and the wolframite rim (cross-polarized light). (**D**) False-colored μXRF map (K·K) of the greisen (sample PNQ-19), imaging the muscovite-quartz assemblage. Note the complicated, lace-shaped morphology of the muscovite flakes and the preferential elongation of the muscovite aggregates along with privileged directions (dotted lines). (**E**) False-colored μXRF map (K·K) of a contact "greisen"-episyenite (sample PNQ-17a). Note the quite sharp "front of episyenitisation" and the apparent preferential orientations of the muscovite aggregates. (**F–G**) False-colored μXRF maps of the greisen/episyenite contact in sample PAN-VI-17 from the L0 level. Note the intimate association between muscovite and wolframite crystals (wolframite pseudo-vein), and the development of micro-vugs in the episyenite body. (**F**) Fe·K map; whatever the location, "greisen" or episyenite, most of the muscovite displays the same iron content, i.e., there is no appearance of compositional zoning, although the Fe-rich fringes are more developed in the episyenite body. (**G**) K·K map; note the quite sharp "front of episyenitisation". (**H**) Photomicrograph of relic magmatic quartz in the L0 greisen (sample PAN-VI-17). Note the embryonic snowball texture. (**I**) SEM-BSE image of compositional and patchy zoning in a muscovite aggregate ("greisenisation") from the G1a granite (sample PAN16-4a). The crystals display a three-zoned texture, with a grey core (Z1), a darker first rim (Z2), and a bright second rim (Z3). Note the Z3 rims and the protolithionite (Prl) relics in one muscovite core.

As at the L2 level, pure muscovite rocks occurred as meter-scale patches (Figure 9A–C). They were here devoid of any feldspar. They were usually full of microvugs, lined with Fe-rich Li-muscovite, and yielding a vacuolar texture to the rock. Wolframite (Frb# around 0.8, rich in Nb, up to 1750 ppm, and Ta, up to 4500 ppm) might be associated with muscovite within such vugs (Figure 9). Late sulfides and carbonates overprinted both the greisen and muscovite assemblages, as at the L2 level.

All the greisenised facies displayed fluid inclusions (FI)—up to several 10 μm in size—which were often decrepitated (Figure 10). The significance of this feature has been examined in a later section of the discussion

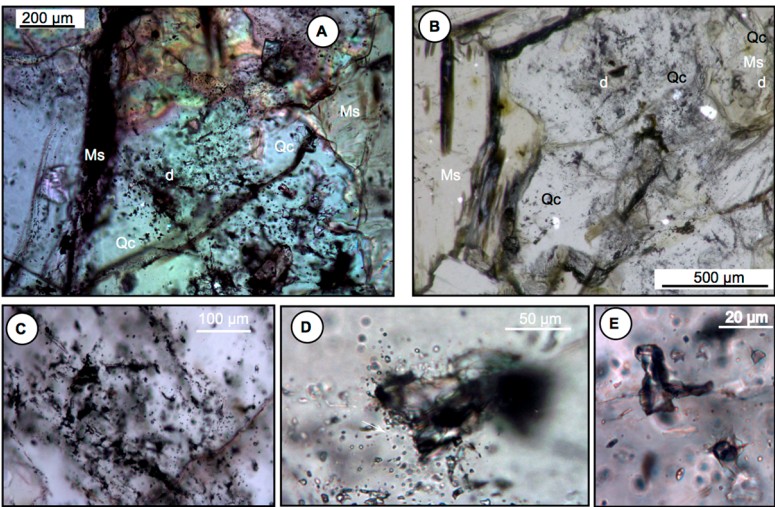

**Figure 10.** Photomicrographs of fluid inclusion (FI) evidence for an early silicification event in the greisenised G4 granite (level L1, sample PAN-XV-4, plane-polarized light). (**A**) Existence of a halo of clear quartz (Qc) around the package of decrepitated FIs (d). (**B**) Clear quartz halo (Qc) isolates the decrepitated FI packages (d) from the surrounding muscovite (Ms). (**C**) Detail of a set of decrepitated FI. (**D**) A large decrepitated FI with a halo of newly formed satellites (arrow). (**E**) Stretched FIs.

*6.2. Petrography: The SCB2 Drill Hole*

Based on macroscopic observations of the drill hole, confirmed by the microscopic study and corroborated by examination of the geochemical profiles, the Panasqueira granite (G1) was subdivided into three facies—G1a, G1b, and G1c from bottom to top (Figure 2C).

6.2.1. Two-Mica Porphyritic Granite (G1a)

The granite is porphyritic, with perthitic Kfs megacrysts up to 6 cm [12] (Figure 11). The Kfs were not densely distributed, with irregular repartition, and they did not display any preferential orientation (e.g., no magmatic foliation). Together with large biotite (up to 5 mm), muscovite (up to 3 mm), albite (up to 1 cm), and quartz (up to 5 mm) crystals, they were set in a fine-grained (~0.5 mm) quartz-albite-muscovite matrix, where albite crystals were either isolated or organized in small clusters. Numerous accessory minerals were present (apatite, ilmenite, rutile, monazite, zircon), apatite being by far the more common, and having the largest sizes (up to 2 mm).

According to [15], the modal composition, based on XRD, is 35% quartz, 28% albite, 17% K-feldspar, 7% muscovite, and 1% apatite. Flakes of muscovite and quartz grains were dispersed within the feldspars, and in particular, in albite (Figure 11C), where a canvas of small muscovite laths might equally be observed (Figure 11A).

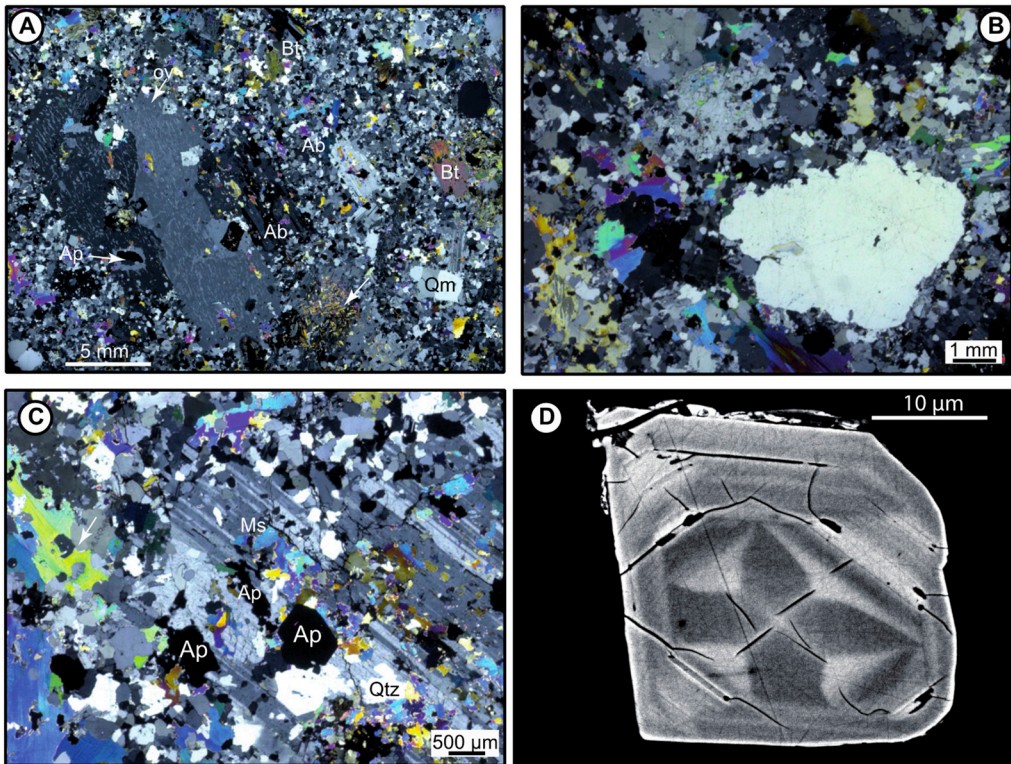

**Figure 11.** G1 granite unit from the SCB2 drill hole. (**A**) Photomicrograph global view of the two mica G1 texture (cross-polarized light, sample PAN-16-5a). Megacrysts (Kfs, Ab, Bt, quartz Qm) are embedded in a fine-grained matrix. Note the perthites in the Kfs and the broad albite rim around the apatite (Ap), included in the Kfs. A rim devoid of perthites (OV) is noticeable in the Kfs. Note also the muscovite canvas in some albite megacrysts (arrow). (**B**) Detail of magmatic quartz in G1, showing lace-shaped overgrowths (r) (microphotograph in cross-polarized light, sample PAN-16-6). Note the amoeboid shape of muscovite crystals. (**C**) Development of patchy quartz (Qtz), muscovite (Ms), and apatite (Ap) in a G1 albite megacryst. Note the independency between quartz and muscovite. The large muscovite flakes display amoeboid shapes (arrow) (microphotograph in cross-polarized light, sample PAN-16-6). (**D**) SEM-BSE image of zoned zircon included in G1 biotite. Note the spectacular sector zoning. Variations in Hf contents mainly control the compositional zoning pattern.

The K-feldspar megacrysts were close in composition to pure orthoclase end-member, with no more than 10% albite component. They typically showed an overgrown fringe (~1 mm thick) devoid of any perthite flames. The perthite component was pure albite. Albite, either as subhedral phenocrysts, or matrix crystals, was equally albite close to An00. Biotite (in fact, a "protolithionite", see below: mineralogy) occurred as subhedral crystal and was usually more or less altered. The alteration was multi-stage. The first transformation occurred along the cleavages of biotite altered into muscovite (Ms0) with the simultaneous development of Mg-bearing siderite (zoned, with an Mg-Mn enriched core). Then, a Fe-chlorite (the chlorite post-dating the siderite) was newly formed. Finally, the second episode of muscovitisation (Ms1) occurred in evident relation with the muscovite flakes outside the protolithionite (Figure 12). The flakes must consequently also be considered Ms1. All transitional states exist between unaltered samples and samples where all biotite is replaced [12]. The first stage (Ms0) was mostly responsible for the biotite replacement, although the final replacement of the early Ms0 by the later muscovite (Ms1) was equally observed.

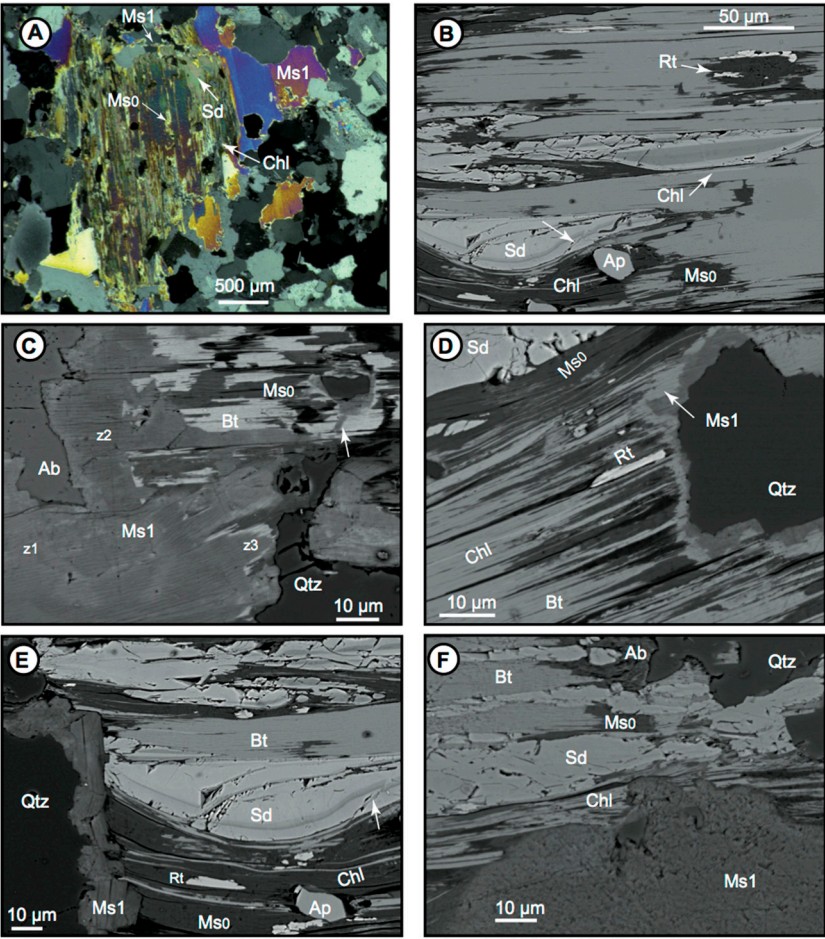

**Figure 12.** Alteration assemblages in G1 biotite (sample PAN-16-5). Ap apatite, Bt Biotite (protolithionite), Chl chlorite, Ms0-Ms1 muscovite, Rt rutile, Sd siderite. (**A**) Microphotograph in cross-polarized light, (**B**–**F**) SEM-BSE images. (**A**) Overview of altered biotite. Note the muscovite microcrack (Ms1) issued from a large Ms1 flake. (**B**) Alteration assemblage. Patchy muscovite (Ms0) is the earlier phase, overprinted by siderite and chlorite. Siderite is overprinted by chlorite (arrow). Note the intimate association between rutile and muscovite Ms0. (**C**) Relationships between a flake of muscovite (Ms1) and the altered biotite (Bt-Ms0 assemblage). Ms1 dilacerates the altered biotite; note in particular the Ms1 microcrack (arrow). Z1-3: zoning of the Ms1 (see Figure 10). (**D**) Overprinting of the Ms0-Chl assemblage by Ms1 developed at a quartz interface. (**E**) Overprinting of the Ms0-Chl-Sd alteration assemblage by an Ms1 microcrack at a quartz-biotite interface. Chlorite overprints siderite (arrow). (**F**) Obliteration of an Ms0-Chl alteration assemblage by an Ms1 flake.

The quartz phenocrysts, being devoid of any concentric albite micro-inclusions, could not be qualified as "snowball quartz", contrary to Launay's statement [42]. Aureoles of quartz digitations around the magmatic quartz were apparent (Figure 11B).

Muscovite phenocrysts and matrix crystals presented all habits between nearly euhedral and extremely amoeboid (Figure 11C), showing a strong compositional zonation pattern in BSE imaging (Figure 12C).

Zircon was euhedral (≤50 μm), mainly included in biotite, and displayed in BSE images spectacular zoning, due to small variations in the Hf contents (from 0.9 wt% $HfO_2$ in the dark cores to 1.5 wt% $HfO_2$ in the clearer bands) (Figure 11D). Biotite, as well as apatite, hosted euhedral monazite (≤50 μm).

Rutile was a secondary mineral, intimately associated with the early Ms0 alteration (Figure 12). It was typically zoned, with a combination of oscillatory (OZ) and sector (SZ) zoning (Figure 13). Ilmenite was a primary mineral, present as sporadic euhedral crystals, but was mainly found pseudomorphosed

by a rare metal-poor rutile, or more commonly, by a symplectic assemblage of rare-metal poor rutile and siderite, suggesting contemporaneity with early muscovitisation (Ms0) of the protolithionite. Apatite formed euhedral crystals in both biotite and the feldspars and disseminated micro-inclusions in the K-feldspar megacrysts. In the K-feldspar megacrysts, apatite was typically embedded into an aureole of pure albite, which was connected to the perthite flames (Figure 11A). Most of the crystals included in either protolithionite or albite were patchy zoned (Figure 14D,E). The largest apatite crystals (1–2 mm) displayed more complicated zoning, with a patchy core and a rim characterized by oscillatory zoning (OZ), and a corroded boundary between them. The core commonly contained micro-inclusions of a variety of phosphates (see below: mineralogy) or Th-rich uraninite (up to 18 wt% $ThO_2$) (Figure 14F,G,K). Zircon (exceptionally, associated with xenotime) was an uncommon companion of these phosphates.

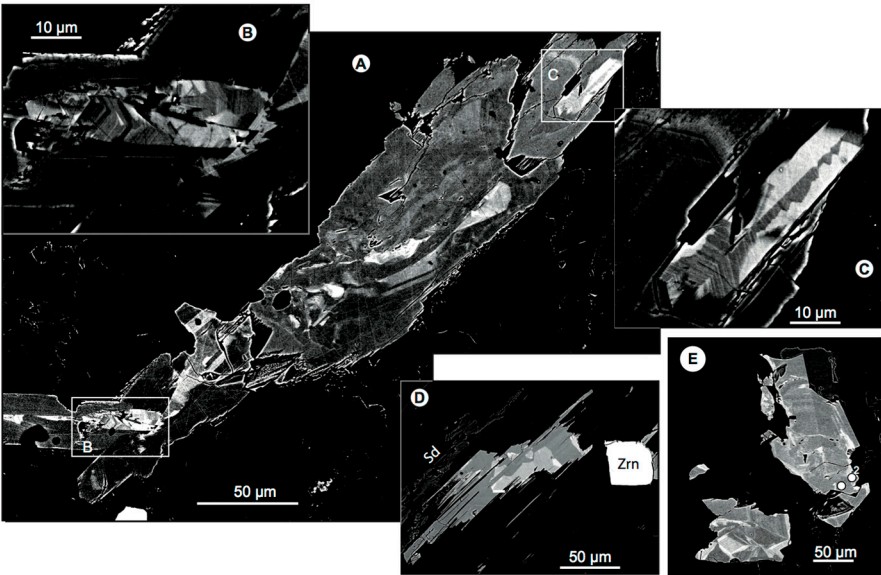

**Figure 13.** Sector zoning in the rutile form from the G1 suite, SEM-BSE images. Note the combination of oscillatory and sector zoning and that prismatic forms are characterized by low endowments in rare metals. (**A**–**C**) Large crystal from sample PAN-16-5b, showing small prisms incorporated in a larger prismatic crystal. (**A**) Overview. (**B**,**C**) details of the smaller prisms. (**D**) Another crystal from the same sample. (**E**) Rutile crystals from the G1a unit (sample PAN16-4). Note the similarity with the G1 rutile.

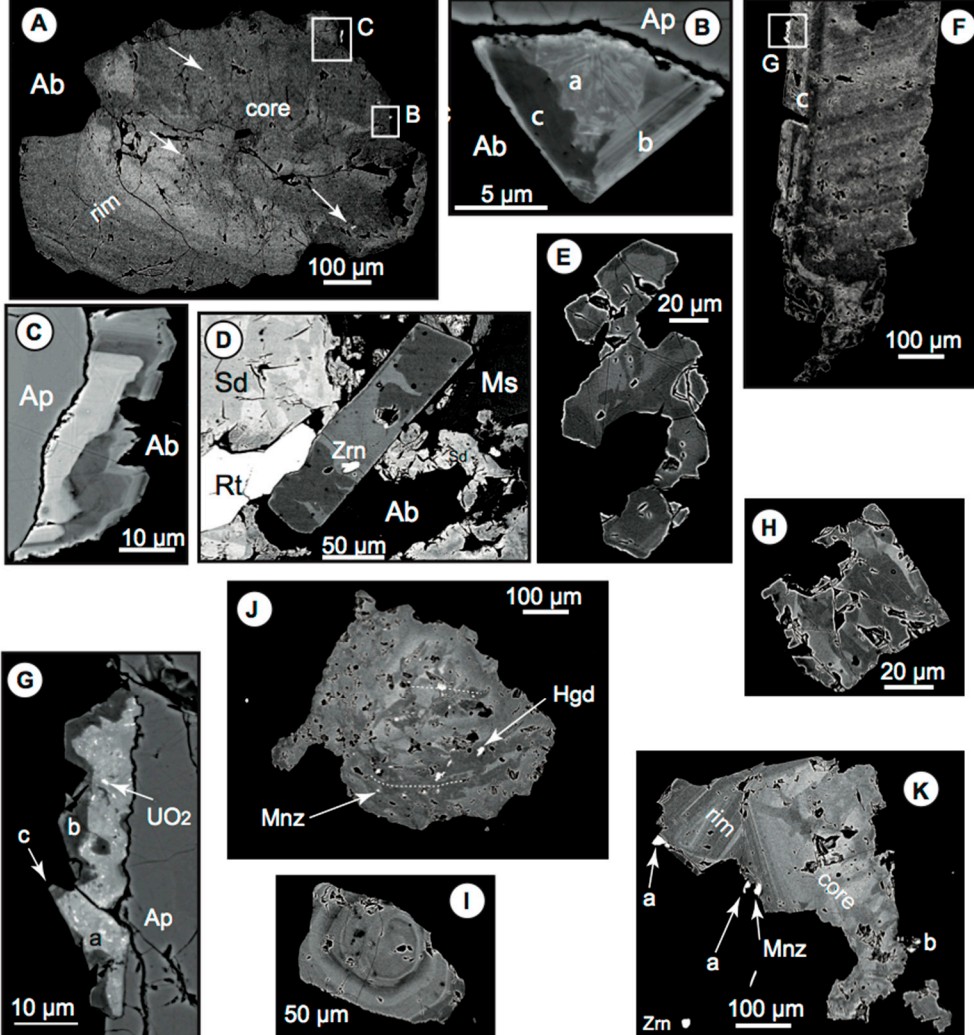

**Figure 14.** Apatite in granites of the G1 suite in the SCB2 drill-hole (SEM-BSE images). (**A–C**) Largely zoned apatite embedded in an albite aureole within a perthitic Kfs from the G1a granite (sample PAN-16-5b, see Figure 10). (**A**) Overview of the crystal, showing a core with inclusions of uraninite, monazite, and Al–Fe–Mn phosphate (arrows), which is corroded/surrounded by a rim with oscillatory zoning (OZ). (**B**) Overgrowth of an alumina-phosphate-sulfate (APS) assemblage; a: fibrous solid solution goyazite-florencite, b: metal-poor sector of a sector zoned crystal, formed by goyazite, c: metal-rich sector, formed by a solid solution goyazite-florencite (OZ is due to variations in Sr and LREE contents). (**C**) Overgrowth of zoned siderite, marked by Mg variations. (**D**) A patchy zoned crystal within an alteration assemblage at the expense of a protolithionite (granite G1a, sample PAN-16-5b). (**E**) Chain of small patchy zoned crystals embedded in an albite crystal included in Kfs (granite G1a, sample PAN-16-6). (**F**) A complexly zoned crystal with an APS overgrowth (**G**) (granite G1b, sample PAN-16-4a). (**G**) Detail of (**F**), showing the APS overgrowth on the apatite; a and c: solid solution goyazite-florencite, b: goyazite; note the numerous uraninite micro-inclusions in a. (**H**) Patchy zoned crystal included in muscovite (granite G1c, sample PAN-16-3a). (**I**) Small zoned crystal (OZ) in the G1c unit (sample PAN-16-4b). (**J**) Large zoned crystal in the G1c unit (sample PAN-16-3b); a patchy core is dotted by micro-inclusions of monazite (Mnz) and hagendorfite (Hgd), with Hgd concentrated in a central domain (dashed line separation). (**K**) Large zoned crystal, included in an albite phenocryst invaded by quartz and muscovite (granite G1c, sample PAN-16-3b); a patchy core is surrounded by a zoned rim (OZ). Overgrowth of monazite (Mnz) and APS in the rim, and the core free of inclusions; a: solid solution goyazite-florencite; b: APS succession and uraninite inclusions as in (**G**).

### 6.2.2. Greisenised Protolithionite Granite Unit (G1b)

The G1b unit differed from G1a by being finer-grained and less porphyritic, with less quartz and albite phenocrysts of a few mm in size and a quartz (0.3–0.5 mm)-albite (0.2–0.3 mm)-K-feldspar (0.2–0.3 mm) matrix (Figure 15). The magmatic quartz displayed an aureole of lace-shaped digitations. It had an embryonic snowball texture, with a few microliths of pure albite (with apatite micro-inclusions) or K-feldspar. The G1b unit was more affected by alteration than G1a. On the one hand, strips of "greisen" with a coarser grain overprinted the rock. On the other hand, late muscovite (Ms1) development was more pronounced (Figure 15). Pristine biotite was practically lacking, saving some minute protolithionite remnants, either preserved in quartz or less commonly present as relicts in secondary muscovite (Ms1). The muscovite presented all habits between subhedral flake (with ragged edges) and amoeboid. It was mainly associated with quartz, but was also seen to overprint albite, in particular, albite phenocrysts (Figure 15B), and was zoned (Figure 6E). Accessory minerals were abundant apatite, with subordinated rutile, and rare zircon.

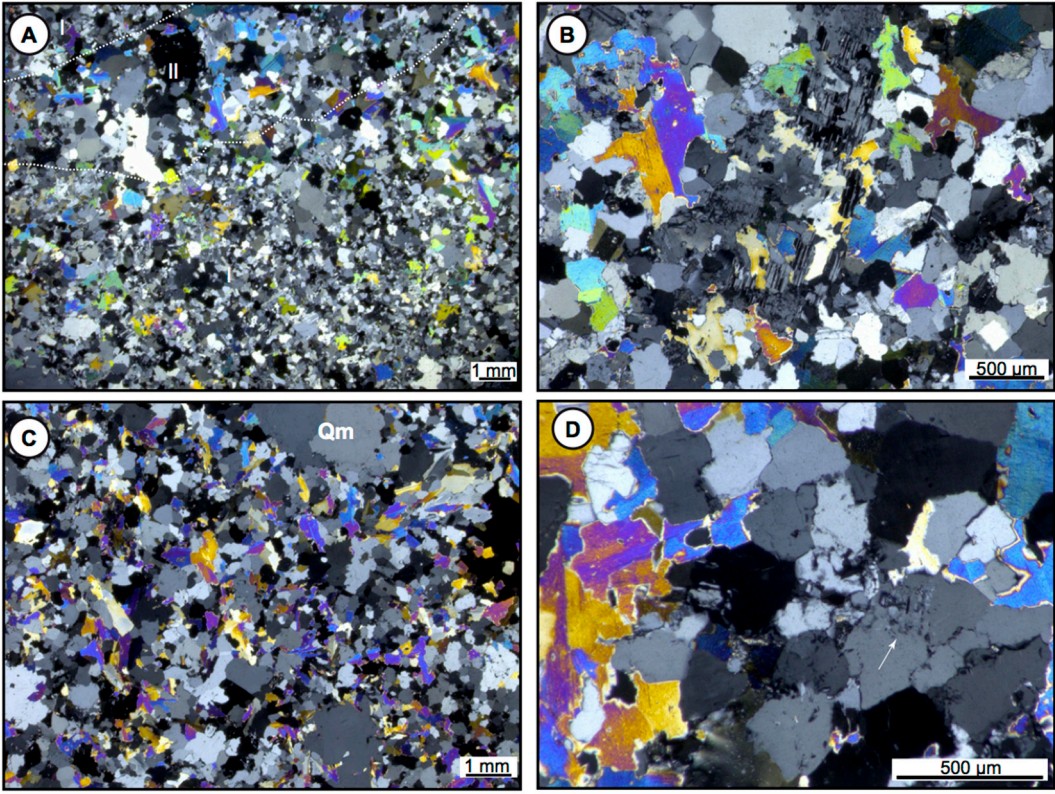

**Figure 15.** "Greisenisation" in the granites from the SCP2 drill-hole (microphotographs cross-polarized light, sample PAN16-4b). (**A**) Partially "greisenised" G1a granite. A strip of a quartz-muscovite assemblage (II) is superimposed on the equigranular albite granite (I). Note the abundance of muscovite in the granite. (**B**) Replacement of a G1a albite megacryst by amoeboid muscovite (sample PAN16-4b). (**C**) Overview of the fully greisenised G1b granite (sample PAN16-3a). Note the preservation of magmatic quartz (Qm). (**D**) Detail of the greisen, showing the relic lace-shaped albite within the quartz (arrow) (sample PAN16-3a).

Apatite occurred as large euhedral to subhedral crystals, commonly included in albite phenocrysts, displaying the same type of zoning as their counterparts in G1a (Figure 14F,G). Oscillatory zoning was rare (Figure 14I).

Most of the rutile crystals displayed the same habit as those issued from the first stage of the alteration of the biotite in G1a (Figure 13E). The associated early muscovite (Ms0) was, however, absent, presumably replaced by late muscovite (Ms1), but, in a few instances, siderite was associated with the

rutile. A few other rutile grains contained micro-inclusions of ilmenite, suggesting the replacement of pristine magmatic ilmenite, as in G1a.

### 6.2.3. Greisenised Protolithionite Granite Unit (G1c)

The G1c unit was a strongly altered facies in the continuity of G1b, e.g., a quartz greisen with patches of a less altered granite. The latter was fine-grained, with a quartz-feldspar (0.2–0.3 mm)-muscovite matrix, with dispersed albite microphenocrysts (0.5–1 mm), larger phenocrysts of albite (≤5 mm), and magmatic quartz (1–3 mm). Feldspars in the matrix occurred either isolated (subhedral) or in anhedral clusters. They were mainly albite, with very few K-feldspar. Magmatic quartz hosted rare protolithionite relicts and presented an embryonic snowball texture with aureoles of digitations. Muscovite occurred as isolated plumose flakes (0.2–0.7 mm), with irregular (more or less ragged) edges. It was mainly associated with quartz, but might commonly obliterate albite (notably, the phenocrysts). Muscovite was zoned (Figure 6H).

The more altered parts of the unit consisted of a quartz-muscovite greisen (Figure 15C), the only remnants of the initial granite being magmatic quartz, numerous lace-shaped ghosts of the former albite (Figure 15D), uncommon topaz ghosts and tiny remnants of a protolithionite, included in muscovite or in quartz. Quartz digitations in the aureole of the magmatic quartz were more important than in the less altered facies. Muscovite crystals occurred as interconnected flakes (up to 1 mm) with a more or less pronounced plumose habit and a tendency to enclose quartz. They formed chains, encompassing magmatic quartz, and locally overprinted the albite ghosts.

Accessory minerals were apatite, minor rutile, and monazite, and rare ilmenite and zircon. A single uraninite crystal (20 μm), associated with a tiny Nb–Ta-rutile (2 μm), was observed in the more altered facies, included in muscovite. The uraninite, devoid of thorium, contained 3.3 wt% Pb.

Apatite was present either as large crystals (up to 500 μm), displaying the same type of zoning as encountered in both G1a and G1b (Figure 14J,K) or as smaller euhedral crystals (≤100 μm) elsewhere (somewhat ragged in the more altered facies), displaying concentric or patchy zoning (Figure 14H). Monazite, as small crystals (up to 50 μm), included in quartz or muscovite, was thorium-rich and displayed concentric zoning due to variations in the Th content. Rutile crystals were small and contained in muscovite. They were unzoned and rare-metal poor. In one occurrence, they could represent a relic of an early muscovitisation of protolithionite (Figure 6H). As in the greisen cupola, late mineral phases, such as various sulfides (arsenopyrite, pyrite, chalcopyrite, sphalerite, galena) and siderite, overprinted all the previous assemblages in the G1c unit.

The G1c unit differed from the subjacent G1a/G1b units by (i) the finer grain, (ii) the presence of topaz, (iii) the scarcity of K-feldspar, and (iv) the lack in Nb-rich rutile. That it pertained to the G1 suite was, however, suggested by (i) the faint porphyritic texture, (ii) the presence of large apatite with other phosphate micro-inclusions, (iii) the presence of a Th-rich accessory phase, and (iv) the textural evidence of a possible stage of early muscovitisation (Ms0) of a protolithionite. It differed from the G2 unit by its finer grain size and its porphyritic texture, and both G2 and G3 units by the lack of columbite-tantalite.

### 6.2.4. Muscovite Leucogranite (G2)

It was an equigranular fine- to medium-grained granite, with a matrix of quartz-albite-(minor K-feldspar)-muscovite (0.5–1.5 mm), and slightly coarser magmatic quartz (~2 mm). Albite was close to An00, as the occasional small albite microliths found in the magmatic quartz (embryonic snowball texture). This facies was free of biotite relics or topaz. Muscovite mainly occurred with quartz, as more or less aggregated flakes (up to 2 mm) with a flexuous-plumose habit, or interconnecting films following quartz grain boundaries, but it also might overprint albite. It was zoned (Figure 6K,L).

Accessory minerals were apatite, zircon, monazite, and columbite-tantalite. Small euhedral apatite crystals (10–20 μm) were abundant and included in albite. They were zoned, either patchy or with a core and a rim (Figure 14E,F), and were frequently associated with zoned siderite (Figure 14D) or

small Sr-rich APS (Figure 14B). A unique occurrence of a larger apatite crystal (~300 μm), with the same zoning features and various phosphate micro-inclusions as in the G1 suite, was included in albite (Figure 14B). Apatite also occurred as pseudo-veinlets, clearly obliterating the muscovite flakes and more or less transformed into an assemblage of Sr-rich APS and kaolinite (Figure 6J,L).

Zircons were uncommon and occurred as tiny crystals (up to 2 μm) included in quartz. They were zoned with variable U content and contained nano-inclusions of uraninite without thorium. Th-rich monazite was locally associated with the small apatite crystals included in albite. Columbite-tantalite formed bundles of small prismatic crystals with conspicuous oscillatory zoning (Figure 16).

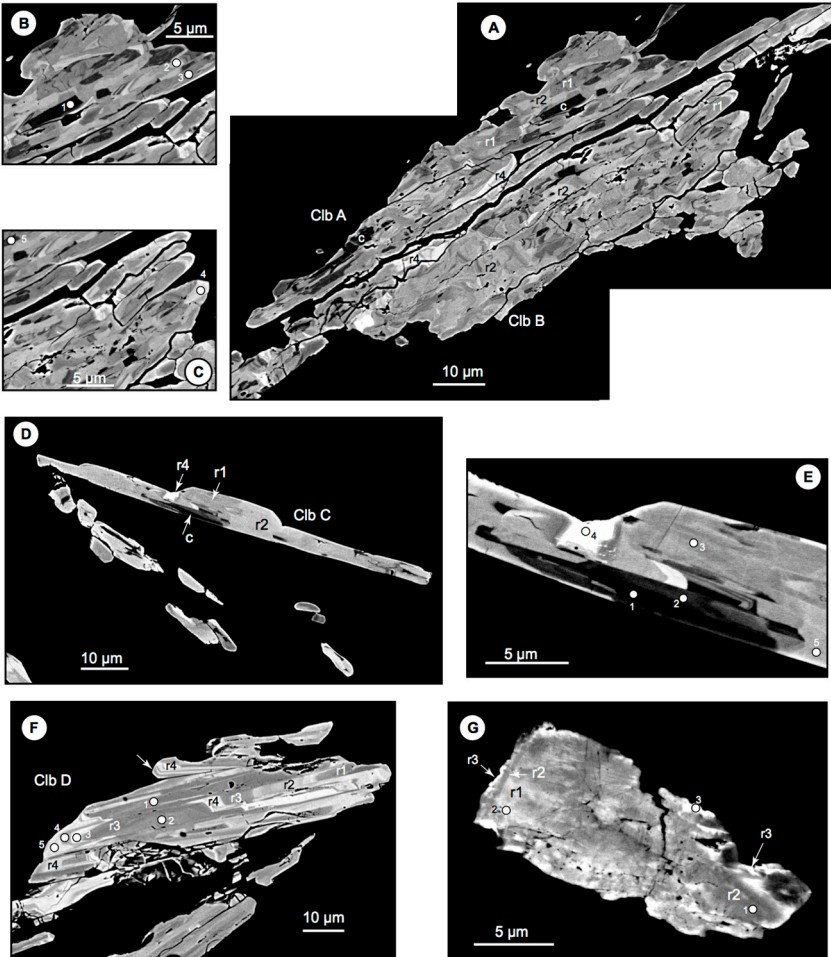

**Figure 16.** SEM-BSE images of the columbite-tantalite (Clb) from the G2 and G3 granite units: c core, r1 to r4 rims (see text). (**A–F**) G2 unit (sample PAN-16-8). (**A**) Two adjoined prisms: Clb A with numerous prismatic dark core sections and the almost full replacement of r1 by r2; by contrast, Clb B is devoid of dark cores, but with the aggregation of r2-r3 sub-units. (**B,C**) details of (**A**), showing the position of analyzed spots (Table S3). (**D**) Acicular Clb prism, displaying replacement of r1 by r2. Note the absence of r3. (**E**) Detail of (**D**), showing the emplacement of analyzed spots. (**F**) Blocky Clb prism with spectacular OZ in r2 and r4 and up to three alternating clear/dark zones in r4 (arrow). The r1 to r4 succession is seen in one of the aggregated prisms. The location of the analyzed spots is indicated. (**G**) Columbite-tantalite from the aplite unit (sample PAN-16-9); r1 to r3 is the proposed succession order. The location of the analyzed spots is shown.

Snowball quartz texture, the lack of Nb-rich rutile, and above all, the abundance of columbite-tantalite marked the difference with the G1 suite, whereas the occasional presence of large apatite with the G1 type of zoning might testify for an affiliation.

### 6.2.5. Muscovite Fine-Grained Leucogranite ("Aplite") (G3)

The G3 facies contrasted macroscopically with the G2 facies by its "aplitic" aspect, as it was a fine-grained leucogranite, with quartz (0.2–0.5 mm)-albite (0.2–0.5 mm)-muscovite matrix, and micro-phenocrysts of quartz and albite (1–2 mm). In the matrix, albite occurred as either subhedral isolated crystals or small anhedral clusters. Albite boundaries were commonly ragged at the contact with quartz. The magmatic quartz had a snowball texture and displayed aureoles of lace-shaped digitations. Biotite or topaz was absent. Muscovite mainly occurred in association with quartz, as more or less aggregated, denticulated to amoeboid flakes (0.2 to 1 mm), interconnected by muscovite films developed at the quartz crystals interfaces. Muscovite was zoned as in other granite facies. Accessory minerals were apatite, zircon, monazite, rutile, and columbite-tantalite.

Apatite was abundant, most commonly included in albite, as small subhedral crystals (10–20 μm). The crystals were zoned, with a corroded core and a rim. Like in G2, zoned siderite patches were commonly associated with apatite in the albite. Locally, tiny albite crystals, with the same composition as the rims, followed healed microcracks in the hosting albite (Figure 7D). Less commonly, larger apatite crystals (200–400 μm) were dispersed in the quartz-albite matrix. Their boundary exhibited a denticulate to lace-shaped habit when in contact with quartz, contrasting with subhedral contours when in contact with albite (Figure 7H,I). When apatite was in contact with muscovite, the latter might display an Mg-rich rim, which formed penetrative indentations into the apatite (Figure 7J). These large crystals were patchily zoned and showed a thin discontinuous rim of a homogenous Mn-poor apatite (Figure 7H,I). They typically enclosed either columbite-tantalite or Nb-rutile (or both) micro-inclusions and might be associated with phosphates of the hagendorfite group (see below: mineralogy). They were permeated by an Mg-siderite (with patches of pure siderite) (Figure 7H,I). Pseudo-veins of the same siderite sinuated throughout the quartz-albite matrix.

Columbite-tantalite was conspicuously zoned, but in style differing from that in G2, with wolframo-ixiolite overgrowths (Figures 16G and 17A–E). It was either dispersed in the large apatite crystals or disseminated in quartz or muscovite. Rutile was always included in apatite, either associated with columbite-tantalite or alone. It was rare metal-rich (see below: mineralogy), but differed by both habit and composition from its counterparts in the G1 suite (Figure 17F–I).

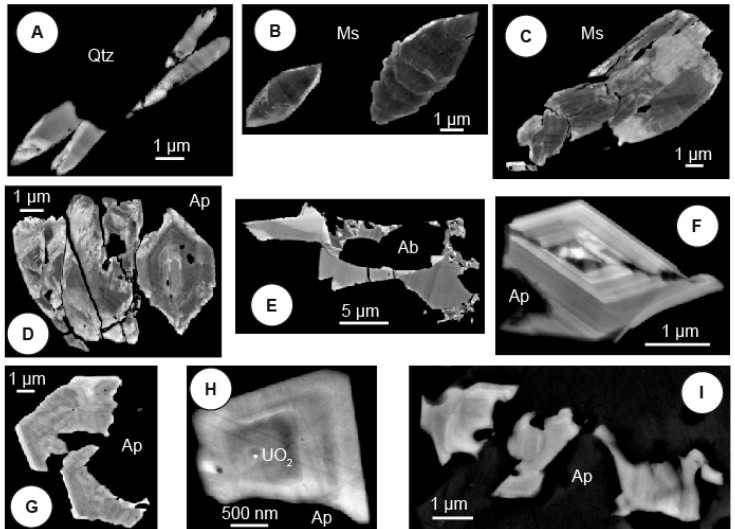

**Figure 17.** Nb–Ta–W oxides in the G3 micro-leucogranite (sample PAN-16-9) (SEM-BSE images). (**A**–**E**) Columbite-tantalite/wolframo-ixiolite. (**A**) Small acicular prisms hosted in quartz (Qtz). Note the OZ and the bright W-rich ixiolite overgrowths. (**B**) Small blocky crystals hosted in muscovite (Ms). Note the conspicuous OZ. (**C**) Blocky prisms in muscovite. Note the importance of the W-rich ixiolite overgrowths. (**D**) Oscillatory zoned crystals hosted in apatite. Note the bright W-rich ixiolite overgrowth and ingress into the OZ crystals. (**E**) The crystal is interstitial in a magmatic albite aggregate. (**F**–**I**) Nb-rich rutile. Note the bright Ta-rich overgrowth in (**G**) and the nano-inclusion of uraninite in the core of the oscillatory zoned crystal in (**H**).

## 7. Results: Mineralogy

### 7.1. Quartz

Cathodoluminescence (CL) images of quartz revealed that, in all the studied rock types, quartz exhibited the same array of CL response, with either blue (Qm), brownish (Q1), or reddish (Q2) luminescence (Figure 18). As the magmatic quartz phenocrysts appeared systematically with blue luminescence (Figure 18A,H,I), it seemed logical to consider all the blue luminescent quartz of magmatic origin. The quartz phenocrysts were zoned, and according to [15], this was due to Ti.

Although all three varieties were present in all rock types, their relative abundance was variable. In the greisen cupola, the bulk of the quartz in the quartz-albite-muscovite matrix was made of the Q1 variety, with cross-cutting Q2 veinlets (Figure 18A–E). By contrast, in the granites from the SCB2 drill hole (from G1a down to the aplite), most of the quartz was magmatic (Q1), the quartz Q2 being minor (Figure 18F–I).

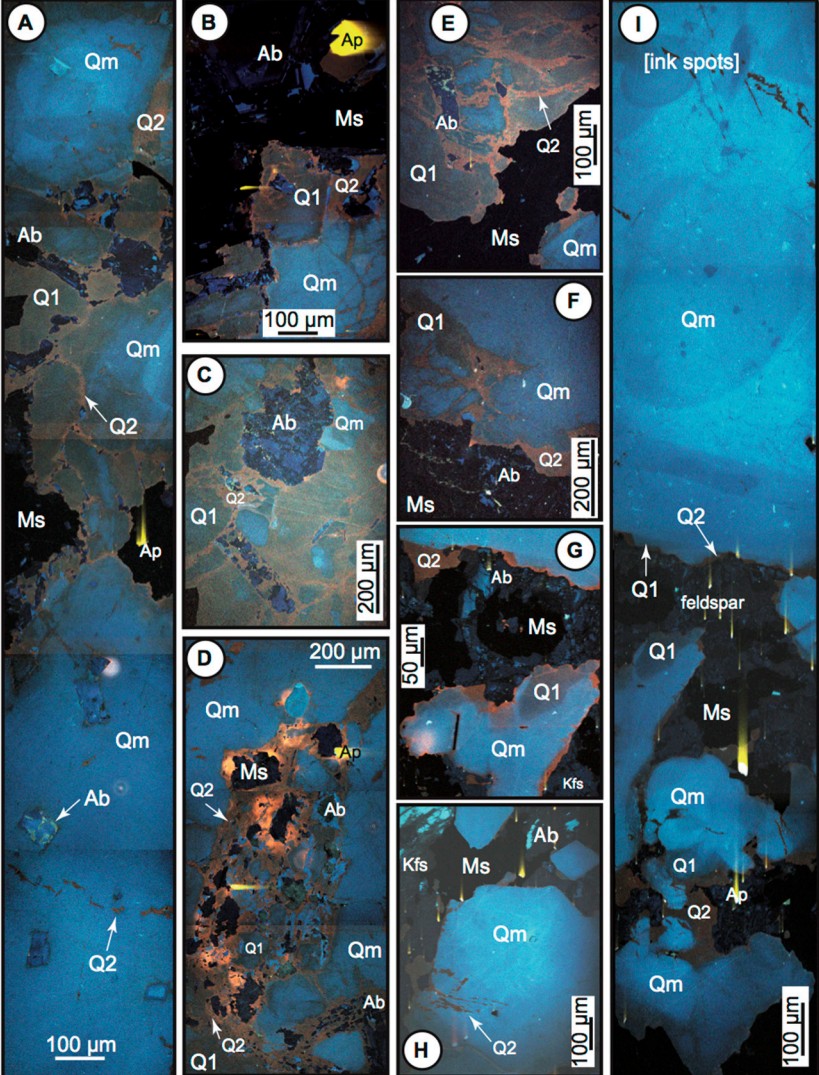

**Figure 18.** Quartz texture revealed by cathodoluminescence (CL, RBG synthetic images). Three luminescence levels discriminate quartz types: blue luminescent quartz (Qm), brownish luminescent quartz (Q1), reddish luminescent quartz (Q2). (**A–E**) Greisenised granite from the L2 level. (**A**) Profile through the contact between a magmatic quartz phenocryst (bottom) and the surrounding quartz-muscovite-albite matrix (sample PNQ-13a). The matrix is constituted of brownish Q1 quartz, crisscrossed by Q2 veinlets. (**B**) Quartz-albite-muscovite of the greisen (sample PNQ-13a). Blue Qm is surrounded by brownish Q1, and both are crisscrossed by Q2. The latter is fringing the muscovite. (**C**) Quartz-albite-muscovite matrix of the greisen (sample PNQ-19a). Cores of blue Qm are isolated within a Q1 matrix, with crisscrossing Q2 veinlets. Albite is corroded and invaded by both Q1 and Q2. (**D**) Quartz-albite-muscovite matrix (Q1 and Q2) in contact with blue Qm (sample PNQ-19a). (**E**) Quartz-albite matrix and muscovite lath (sample PNQ-19a). Q1 surrounds Qm cores, and Q2 veinlets crisscross both. (**F**) quartz phenocryst border in a granite G2 from the SCB2 drill hole (sample PAN16-8). Q1 and Q2 dilacerate the blue magmatic quartz Qm. (**G–I**) porphyritic granite G1 from the SCB2 drill hole. (**G**) Quartz-feldspar-muscovite matrix of the porphyritic granite (sample PAN16-7a). Quartz is mostly formed by blue luminescent Qm, associated with minor Q1, and fringed by reddish luminescent Q2. (**H**) Zoned magmatic quartz grain included in a perthitic K-feldspar megacryst (sample PAN16-5b). Note the Q2 expansions into the K-feldspar and as microcracks within the magmatic quartz. (**I**) Profile from a zoned quartz phenocryst to the quartz-muscovite-albite-Kfs matrix (sample PAN16-7a). The great majority of the quartz consists of blue (magmatic) Qm, with tiny and discontinuous fringes of Q1 and Q2.

### 7.2. Feldspar Microliths in Snowball Quartz

In granites with magmatic quartz characterized by an embryonic snowball texture, the albite and K-feldspar microliths were nearly pure end-members in composition. Only traces of phosphorus could be detected in one K-feldspar microlith from the G3 unit.

By contrast, the albite microliths in the typical snowball quartz from the greisenised G4 cupola commonly contained minor Ca and K, whereas the K-feldspar microliths contained a few Na. Moreover, they were both commonly enriched in P, from 0.4 to 1.0 wt% P in albite, and 0.4 to 1.8 wt% P in K-feldspar. The average structural formula were thus $(Na_{0.92}Ca_{0.02}K_{0.02})(Al_{1.03}P_{0.04}Si_{2.94}O_8)$ for the albite microliths and $(K_{0.95}Na_{0.02})(Al_{1.04}P_{0.04}Si_{2.93}O_8)$ to $(K_{0.96}Na_{0.02})(Al_{1.07}P_{0.06}Si_{2.87}O_8)$ for the K-feldspar microliths.

### 7.3. Protolithionite and Zinnwaldite

In the Monier and Robert triangular diagram ($Al–Si–R2^+$) [52] for mica characterization, the trioctahedral micas from the porphyritic granite G1 plotted in an intermediate position between an annite-siderophyllite solid solution and zinnwaldite (Figure 19A). Such compositions were traditionally named "protolithionite" (e.g., [52]), and, although disregarded by IMA, we have kept it in the following. Plotted in the same diagram, the biotite relicts found in other granite facies (G1a, G1b, G3) appeared to equally have protolithionite, indiscernible from the G1 protolithionite (Figure 19A).

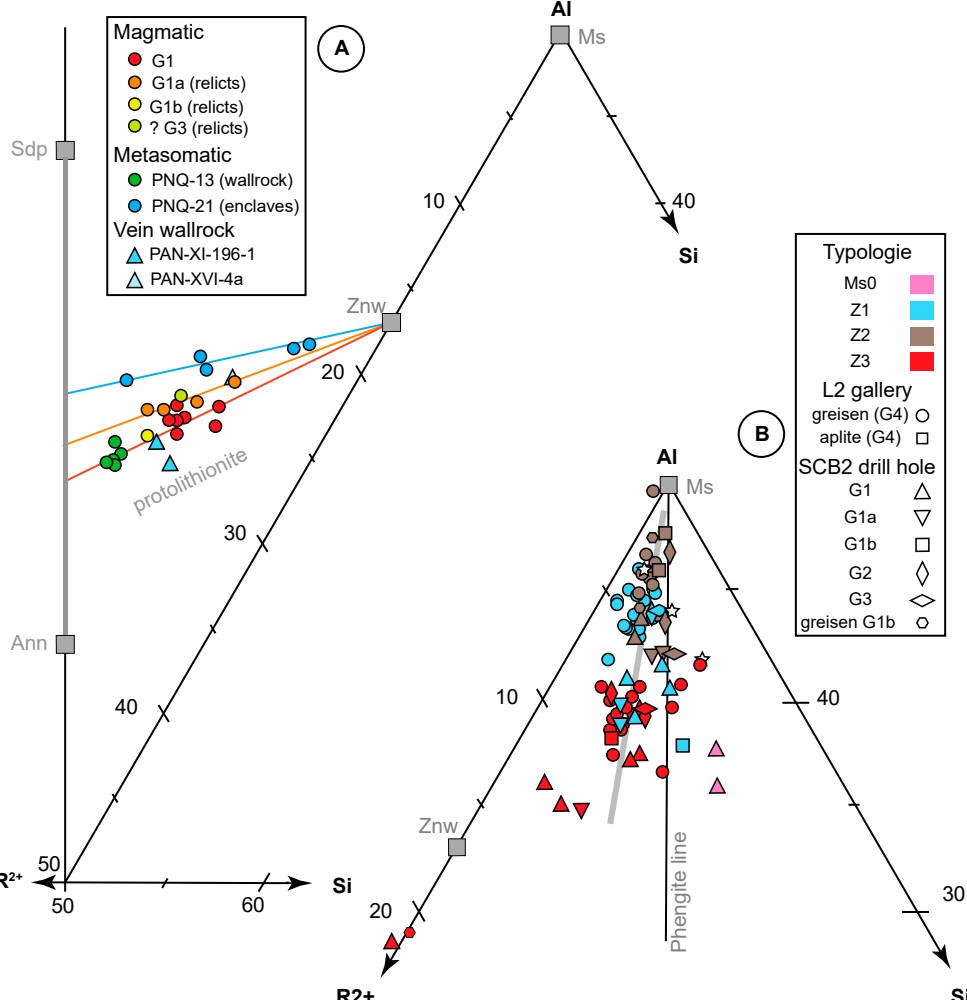

**Figure 19.** Mica chemistry in the $Al-R^{2+}-Si$ diagram [51]. (**A**) Trioctahedral micas; (**B**) Dioctahedral micas. Ms = muscovite, Sdp = siderophyllite, Ann = annite; Znw = zinnwaldite. Legend of symbols, see text.

The metasomatic micas invading the wall-rock of the greisen cupola (Figure 4) were equally of protolithionite composition, with, however, a higher Mg# averaging 0.40 (Figure 19A) (Table S1). This higher Mg# was similar to the 0.40 Mg# of the Beira schists [17]. By contrast, according to their projection in the Monier and Robert diagram [51], the small trioctahedral micas accompanying the muscovite in the quartz enclaves from the G4 granite at the L2 level (Figure 7) appeared to be zinnwaldite, with practically no Mg (Figure 19A).

### 7.4. Li- and Fe–Li-Muscovite

In all muscovite crystals from all granite types, greisen, and muscovite, compositional zoning, with basically the same characteristics from either the textural or the chemical composition point of view, was observed.

### 7.4.1. Muscovite Zoning

In the BSE images in Figure 6, the compositional zoning consisted of three zones (Z1 to Z3), differing by their grey hues, increasingly lighter in the Z2-Z1-Z3 order. The basic pattern was more or less concentric, with a Z1 core, a Z2 rim, and a Z3 fringe (Figure 6G), suggestive of a Z1, Z2, and then Z3 order of apparition in the growth process. There was a considerable variation; many of the studied crystals exhibiting more or less patchy zoning, leading to complex relationships between the zones. The Z1 zone was, however, systematically overprinted by the Z2 one, and both were overprinted by Z3 (Figure 6C–E). On the other hand, the presence of a Z3 rim was nearly systematic. In most greisenised facies and saved rare crystal relicts (as in Figure 9I), the previous occurrence of a protolithionite or another primary Li-mica was not recorded in the muscovite, to a few exceptions, shown in Figure 6F–I. The slightly Fe-enriched and patchy zoned core of some crystals might thus represent the ghosts of a previously altered Fe–Li-mica. In this perspective, the occurrence illustrated in Figure 6H, where such a core was associated with a few small rutile prisms, might be considered as particularly significant.

### 7.4.2. Muscovite Composition

In the Monier and Robert diagram [51], most of the muscovite crystals from the Panasqueira granites, greisen, and muscovite plotted along a trend diverging from the phengite line, meaning that the micas must be classified as Li-muscovite to Fe–Li-muscovite (Figure 19B) (Table S1). The fields for the three Z1 to Z3 zones were on the whole well-defined, although there were some degrees of recovery. What was more, these fields were the same, whatever the granite or the intensity of its alteration. Consequently, the crystallization of muscovite might be taken as a unique phenomenon at the scale of the Panasqueira granite body, or, at least, of the upper part of the body, in the influence zone of the tungsten deposit (Figure 2A).

### 7.5. Apatite and Associated Phosphates

### 7.5.1. Apatite

Apatite was everywhere fluorapatite with variable Mn and Fe contents (up to 5.0 wt% MnO, up to 1.5 wt% FeO), and as a rule, Mn is more abundant than Fe (Table S2). Strontium was encountered in some apatite crystals from the G1 suite, with contents between 0.3 and 1.4 wt% SrO (up to 2.7 wt%) and from the G4 greisens (1.2–1.3 wt% SrO), without particular relationships with the zoning features.

It was nevertheless not a systematic component of the Panasqueira granites apatite. When the apatite crystals were simply zoned, with a core and a rim (in many cases, with a patchy habit of the core, see Figures 7 and 14), the core was enriched in Mn and Fe, whereas the rim was close to the pure F-apatite end-member (Figure 20). In the case of the large crystals with a patchy core and a homogenous (or OZ) rim (Figures 7 and 14), the highest Mn and Fe contents were systematically found in cores. In contrast, the rims displayed contents similar to those of the simply zoned crystals (Figure 20). There was a tendency to be more and more enriched in Mn when going from the G1 suite

to the G2-G3 group and finally to the G4 greisens (Figure 20). The regular presence of a series of microinclusions in the cores characterized the large apatite crystals with patchy cores. These inclusions consisted of a variety of phosphates, namely, monazite, xenotime (very rare and associated with the zircon), and Fe–Mn–(Mg) phosphates of the alluaudite and wagnerite groups, as well as a few zircons and rare uraninites.

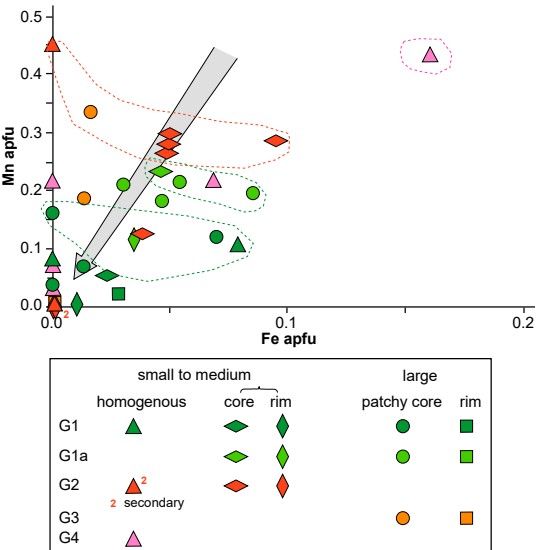

**Figure 20.** Mn vs. Fe diagram for Panasqueira apatite. Zoning is due to limited variations in the iron and manganese contents, with a tendency to decrease from cores to rim in the crystals. The dashed lines domains correspond to interpreted magmatic fields (pink: G4; red: G2-G3; light green G1a; green: G1). The grey arrow materializes the overall trend of subsolidus reworking towards compositions closer and closer to the pure F-apatite end-member. See the text for explanations.

### 7.5.2. Monazite

Monazite was the most common of the microinclusions, together with the phosphates of the alluaudite group. It might be classified as a Ce-monazite and was commonly Th-rich (Table S2).

### 7.5.3. Alluaudite Group

The structural formula of the alluaudite group is given as $(2)'A(1)A(1)M(2)M2(PO_4)_3$, where $(2)M$ is filled with $R^{3+}$, Mg, and $Fe^{2+}$, $(1)M$ with $Fe^{3+}$, $Fe^{2+}$, and Mn, $(1)A$ with $Fe^{2+}$, $Mn^{2+}$, Ca, and Na, and $(2)'A$ with Na and K (commonly, with a lacuna □) [53]. In the structural formulas of the Panasqueira phosphates of the alluaudite group, calculated on a 3P basis, $Fe^{3+}$ (estimated by charge balance) is comprised between 0.33 and 1.19 apfu, and the $Fe^{2+}$ content in the $(2)M$ site is always higher than 0.5. These phosphates may be thus classified as members of the hagendorfite family [53]. In a general way, hagendorfite from the G1 suite differed from all the others by containing Mg (mainly, in G1) and being less endowed in Mn (Table S2). More precisely, the crystals included in the G1 apatite appeared as ferro-hagendorfite, with structural formulas like $(Na_{0.87}K_{0.25})$ $(Na_{0.13}Ca_{0.25}Mn_{0.62})$ $(Fe^{2+}_{1.07})(Fe^{2+}_{1.33}Mg_{0.44}Fe^{3+}_{0.33})(PO_4)_3F_{0.32}$, whereas the true hagendorfite, with formulas, for instance, $(Na_{0.54}K_{0.14}\square_{0.32})$ $(Na_{0.37}Ca_{0.17}Mn_{0.46})(Mn_{0.56}Fe^{2+}_{0.44})(Fe^{2+}_{1.00})(Fe^{2+}_{0.81}Fe^{3+}_{1.19})(PO_4)_3$, were typical of the G2 to G4 suite. The less common micro-inclusions of the wagnerite group were poor in magnesium and by far richer in manganese than in iron (Table S2). They must, therefore, be classified as zwieselite [54], with formulas like, for instance, $(Ca_{0.02}Mn_{1.37}Fe_{0.58})(PO_4)F_{1.23}$. Another characteristic of the large complexly zoned apatite crystals was the presence of overgrowths characterized by tiny crystals of alumina-phosphate-sulfate (APS) minerals (Figure 14). As seen in Figure 21, these minerals were complex solid solutions between Ca, Sr, and LREE end-members, with a dominant of goyazite-svanbergite ± florencite mixtures, for instance, $(Ca_{0.13}Sr_{0.50})$ $(La_{0.06}Ce_{0.14}Pr_{0.01}Nd_{0.04})(Al_{2.90})$

$(P_{1.97}O_4)(S_{0.20}O_4)$ and $(Ca_{0.11}Sr_{0.94})(Al_{2.86})$ $[P_{1.50}O_3(OH_{0.67}F_{0.34})](S_{0.59}O_4)$. In the florencite component, Ce was the dominant REE, as in the case of monazite micro-inclusions in the apatite.

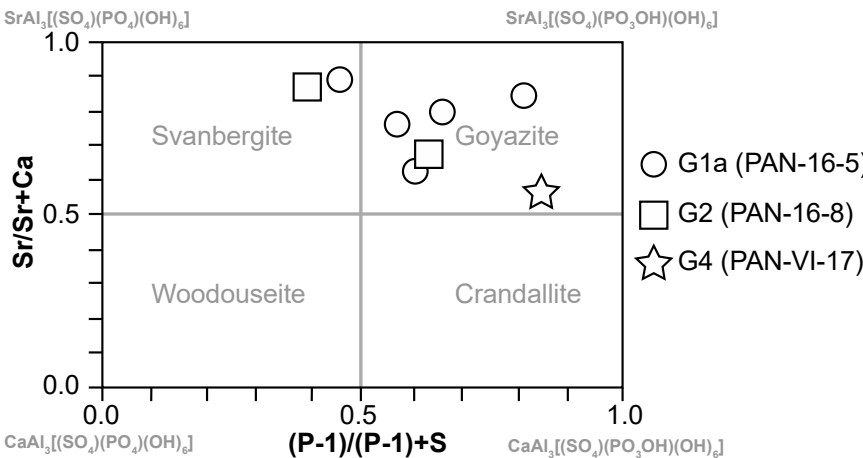

**Figure 21.** Characterization of the alumina-phosphate-sulfates (APS) associated with apatite in Panasqueira granites (following Salama 2014). The quadrilateral S/Sr + Ca versus (P-1)/(P-1) + S diagram of the beudantite and crandallite group minerals shows that Panasqueira APS is a svanbergite/goyazite solid solution (with minor woodhouseite and crandallite components).

*7.6. Nb–Ta–Ti-Oxides*

7.6.1. Columbite-Tantalite

Columbite-tantalite was only present in the G2 and G3 units and was not observed either in granites from the G1 suite or in the greisenised G4 granite, although in the latter, it might have been dissolved, as discussed below. In the G2 granite, the more or less acicular prisms of columbite-tantalite, a few 10 μm to 100 μm and more in length, were displayed with BSE imaging spectacular zoning, combining features of domain (or patchy) zoning and oscillatory zoning (OZ) (Figure 16). The textures were complex, characterized by cracks and replacement features. The larger crystals appeared as the aggregation of several sub-units, as in Figure 16A,F. The textures might be interpreted as resulting from the successive crystallization of a very dark core (c in Figure 16) and up to four rims of variable shades of grey (r1 to r4 in Figure 16), alternating darker (r1, r3) and clearer (r3, r4) zones. Complete r1 to r4 sequence is shown in Figure 16F. Each rim might itself display OZ, as in Figure 16E (r1),A (r2), F (r2 and r4). The sequence was, however, always incomplete. The overall G2 columbite-tantalite composition was Nb- and Fe-rich, with a Ta# (Ta/Ta + Nb) number between 0.05 and 0.25, and a Mn# (Mn/Fe + Mn) number between 0.07 and 0.20 (Table S3). In the classical Ta# vs. Mn# diagram, the compositional zoning was expressed as back and forth evolution along a trend characterized by Ta# increase correlated with Mn# decrease (or constancy) when going from dark to clear zones (Figure 22).

In the fine-grained leucogranite ("aplite") G3, the Nb–Ta oxides presented a more blocky habit and were smaller (15–20 μm). In BSE images, they displayed a less conspicuous oscillatory zoning, with alternating darker and clearer zones, and a bright external rim (Figure 17A–E). By contrast with the G2 granite, where the central zones of the crystals had typical columbite-tantalite compositions, the bright external rim was a wolframo-ixiolite, and most of the clear zones were solid solutions of wolframo-ixiolite with columbite-tantalite (Figure 23). This extreme tungsten enrichment in the late evolution of the Nb–Ta oxides was a characteristic of the G3 granite (Table S3). Besides, they were as an average slightly enriched in Mn (Mn# from 0.20 to 0.27) when compared with the G2 columbite-tantalite, although displaying the same range of Ta# and the same back and forth evolution trend.

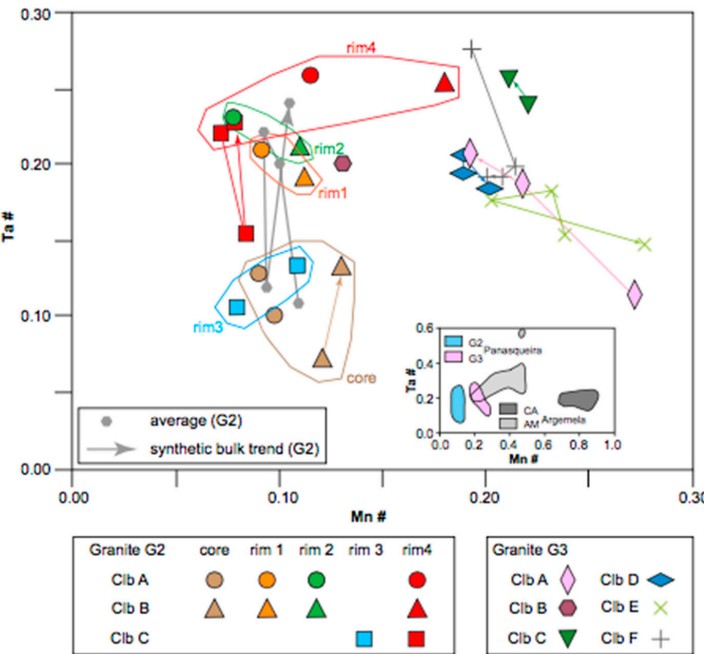

**Figure 22.** Panasqueira columbite-tantalite in the Ta/Ta + Nb (=Ta#) vs. Mn/Fe + Mn (=Mn#) classification diagram. Inset: comparison with the magmatic columbite-tantalite from the nearby Argemela RMG (CT1 [55]: CA, Cabeço de Argemela cupola; AM, Argemela Mine RMG, and dike).

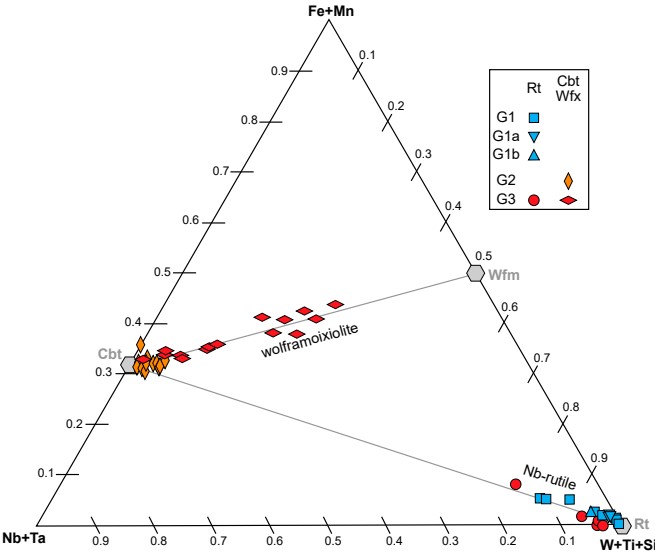

**Figure 23.** Panasqueira colombite-tantalite, wolframo-ixiolite, and rutile compositions in the Nb + Ta/Fe + Mn/W + Ti + Si triangular diagram.

7.6.2. Rutile

Large abundant crystals occurred in the G1 suite, where they were the only bearers of rare metals. They were secondary, resulting from the replacement of protolithionite by muscovite (Ms0). The crystals, up to 200 μm in size, displayed by BSE imaging a conspicuous sector zoning (SZ), combined with oscillatory zoning (OZ) (Figure 13). As well seen in Figure 13, the metal-depleted sectors corresponded to prism faces and the metal-enriched sectors to pyramid forms, in good agreement with the findings of [45]. Prisms were far more developed than pyramids, and as a result, the average rare metal content of the rutile remained low, close to the prism content. In the G1a unit, besides a consistent ~1 wt% Fe (prism) to 3.4 wt% Fe (pyramid), the concentrated metals were V, Nb, Ta, Sn, and W (Table S3). The depleted prism sectors were poor in W (from below detection level (bdl) up to

0.55 wt% W) and contained more Nb (from bdl up to 3.1 wt% Nb) than Ta (0.1 to 0.7 wt% Ta). In the enriched pyramid sectors, W enrichment (up to 1.2 wt% W) was less pronounced than for Nb (up to 10.1 wt% Nb) and for Ta (up to 4.0 wt% Ta). Both Sn (0.4 to 1.8 wt% Sn) and V (0.6 to 1.9 wt% V) were less variable. Whereas rutile was absent from the G2 facies, rare-metal bearing rutile was found in the G3 leucogranite, differing, however, from its counterpart in the G1 suite by the host (apatite), the habit (Figure 17, to be compared with Figure 13), and the common association with columbite-tantalite (Table S3).

## 8. Results: Bulk Rock Geochemistry

The analyses carried out for the present work are given in (Tables S4 and S5). They were used together with data collected from previous works [15–18,56].

### 8.1. Major Elements

Granites and the greisens showed several significant differences between them, greisens being mostly derived from the alteration of the G4 unit. Both of them were highly siliceous, with $SiO_2$ contents between 72.47 and 75.03 wt% $SiO_2$ in the granites and between 70.10 and 80.70 wt% $SiO_2$ in the greisens (whereas the episyenite was no more than 50.57 wt% $SiO_2$) (Table S4). There was a large range of $K_2O$ to $Na_2O$ ratios, the granites ranging from rather albitic to mildly potassic ($K_2O$ to $Na_2O$ ratios between 0.46 and 3.12). In contrast, the greisens, with $K_2O$ to $Na_2O$ ratios between 1.81 and 26.00, were moderately-to-strongly depleted in sodium (Figure 24A), in good agreement with the petrographic evidence. All rocks were rather poor in calcium, the CaO content being consistently ≤0.9 wt% (up to 1.90 wt% CaO in the episyenite, however). They were significantly enriched in phosphorous, with $P_2O_5$ contents in the 0.18 to 0.92 wt% range (up to 1.45 wt% $P_2O_5$ in the episyenite). There was a strong correlation between phosphorous and calcium content, meaning they were both entirely contained in apatite (Figure 24B). The exception concerned a few granite samples of the G1 suite, in which a small excess in calcium was evident due to the calcic cores of the plagioclase macrocrysts (see petrography section). There was otherwise a clear trend of calcium and phosphorous decrease (i.e., a lesser content in apatite) in the greisens relative to the granites (Figure 24B). Mafic components were low, the FeO+MgO+$TiO_2$ sum being comprised between 1.71 and 3.12 wt% in the granites, and slightly higher in the greisens (1.80 to 5.39 wt%, with an outlier at 14.04 wt%), the highest values in the latter being, however, mostly due to a sulfide component (mainly, pyrite and arsenopyrite, with S and As contents of several 1000 ppm) in the mineral assemblage. All the rocks are magnesium poor, with the M index [57] (M = 100 Mg/Mg + ∑Fe) between 3 and 31, higher (M from 15 to 31) in the granites than in the greisens (M from 3 to 15).

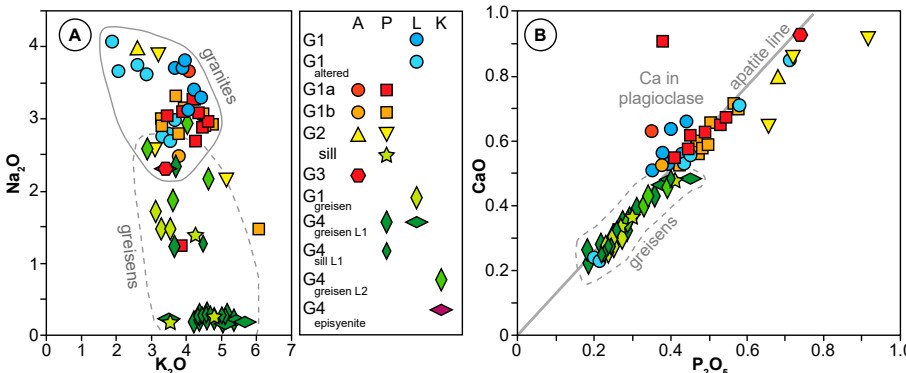

**Figure 24.** Major element geochemistry of the Panasqueira granite suite. (**A**) $Na_2O$ vs. $K_2O$ diagram, showing the sodium leaching in the greisens. (**B**) CaO vs. $P_2O_5$ diagram, showing that calcium is essentially sequestered in apatite. Note that there is less apatite in greisens than in granites. A: [17], P [16], L [15], K [18].

All the rocks were distinctly peraluminous, as seen in the Al-(K + Na + 2Ca) vs. Fe + Mg + Ti diagram of [58] (Figure 25A). The peraluminous parameter Al-(K + Na + 2Ca) was, however, increased by alteration effects, very obvious for the most altered facies, but also perceptible for many of the less altered granite facies. This increase resulted in a blurring of the primary magmatic relationships, the only evident trend in Figure 25A being a "greisen trend". Any attempt to disentangle alteration and magmatic effects must await the discussion below.

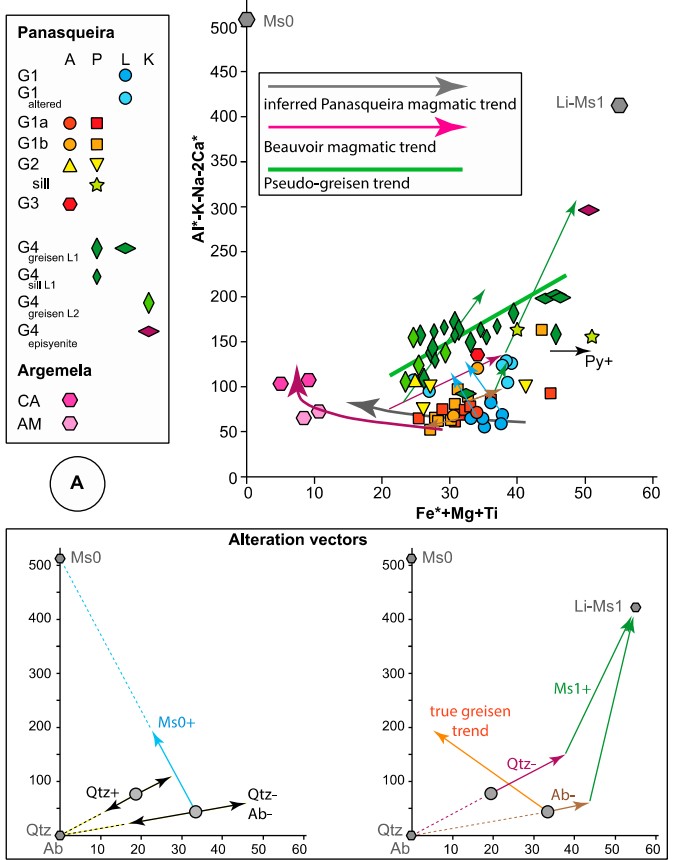

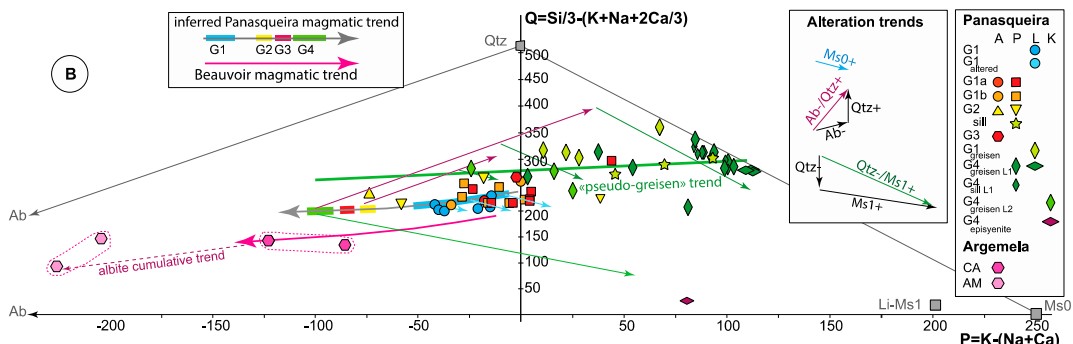

**Figure 25.** *Cont.*

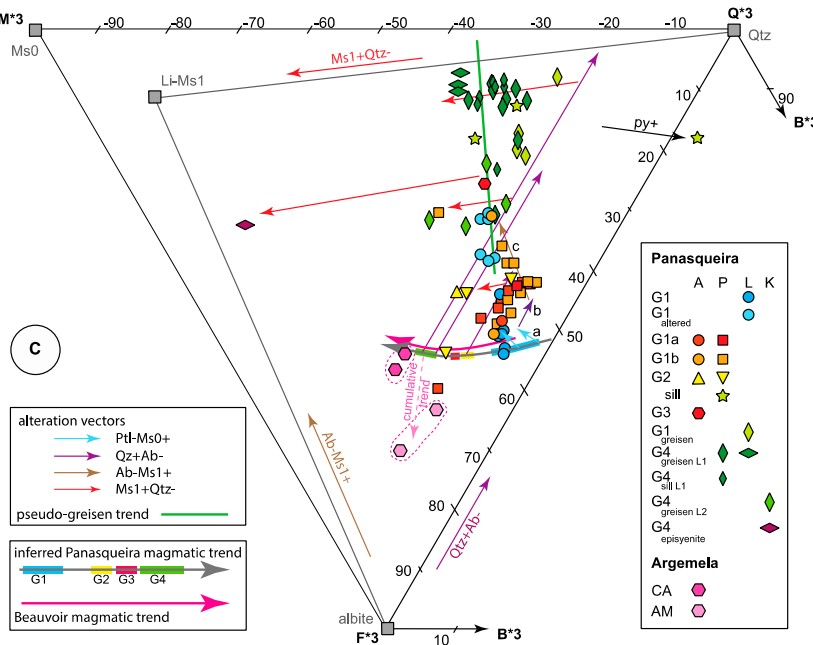

**Figure 25.** Disentangling magmatic and alteration trends of the Panasqueira granites and greisens in the appropriate chemical-mineralogical diagrams. The nearby Argemela RMG and the Beauvoir RMG trend are plotted in the same diagrams as a reference. Data from [15–18,57,58]. Inserts: definition of alteration vectors. Ms0: early muscovite; Li-Ms1: main muscovite; Ms0+ vector: alteration vector corresponding to the early muscovitisation (Ms0 growth); Ms1+ vector: alteration vector corresponding to main muscovitisation (Ms1 growth); Qtz+: alteration vector corresponding to quartzification; Qtz-: alteration vector corresponding to quartz dissolution: Ab-: alteration vector corresponding to albite dissolution. For the construction of all diagrams, the parameters are corrected to take into account both the calcium excess in apatite (for AB and QP diagrams) and the iron excess due to the presence of sulfur (for all diagrams), at least when sulfur analyses are at hand (see Table S2). Such iron excess is still visible for some analyses plotted in the different diagrams, where it is signaled by a "pyrite +" vector. (**A**) Interpretation of the Panasqueira granites and greisens compositions in the A-B diagram (see File S1 for explanations). In this diagram, where quartz and the feldspars project at the origin, the A parameter is set to represent peraluminosity, whereas the B parameter corresponds to the mafic components. In this diagram, it is much easier to check the effects of muscovite on the alteration trends. It appears that the succession of quartzification and muscovitisation in the G2-G4 suite conducts to a bulk alteration trend, nearly orthogonal to the real greizenisation trends (pseudo-greisen trend), whereas in the G1 suite, the effects of the early muscovitisation (Ms0) are predominant. Inversing the alteration trends allows us to reconstruct a pristine Panasqueira magmatic trend, which appears to be very similar to those of typical RMG of the Beauvoir type. (**B**) Interpretation of the Panasqueira granites and greisens compositions in a QP diagram (see File S1 for explanations). In this diagram, it is particularly evident that the G1 suite is poorly affected by the alterations affecting the G2-G4 suite. Owing to the limited effects of early muscovitisation, the G1 suite still displays a magmatic evolution trend very similar to the less evolved part of the Beauvoir RMG trend. Inversing the process in the greisens allows a prolongation of the trend towards the G2-G4 sequence. (**suite**) (**C**) Interpretation of the Panasqueira granites and greisens compositions in the Q*3B*3F*3 diagram. In this double triangular diagram, with parameters Q*3 expressing quartz, and F*3, the feldspars, B*3 opposes mafic minerals (the positive values), and peraluminous components (the negative values, expressed as M*3) (see File S1 for explanations). In this diagram, in the G1 suite, the albite transformation into Li-Ms1 and a limited quartzification become sensitive, both alterations having succeeded in the early muscovitisation (Ms0) (the a-b-c sequence in the diagram). As a consequence, the corresponding alteration trends are aggregated, with the pseudo-greisen trend still prevailing in the G2-G4 suite. Inversing the process allows us to reconstruct the Panasqueira magmatic trend, which, here as in A and B, appears similar to the classical Beauvoir trend.

### 8.2. Trace Elements

As a whole, Panasqueira granites and greisens displayed the trace element enrichment (Rb, Cs, Nb, Ta, W, Sn, U) or depletion (Ni, V, Sc, Th) characteristic of more or less evolved leucogranites (Figure 26) (Table S5). In Figure 26, significant differences in the trace element contents were evident between the SCB2 granites and the cupola greisens, which seemed related either to a more evolved nature of the G4 unit (rare metals, Hf, Cr, Sr, . . . ) or to the hydrothermal alteration (As, Cu, Pb, Zn, Cd, Bi, . . . ).

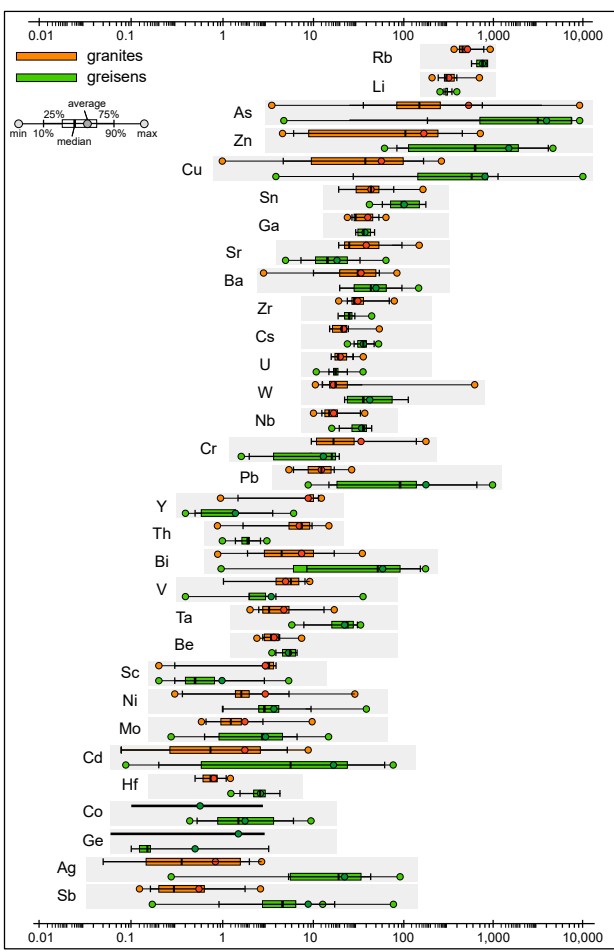

**Figure 26.** Box-plot of minor and trace element distribution in the Panasqueira granites and greisens (in ppm, log. scale).

The different granite units of the Panasqueira suite were discriminated against their trace elements, as clearly seen in the SCB2 drill hole profiles in Figure 27 (Table S5). The contrast was maximal between G2 and G1a units, with distinctive jumps in the Th, Ti, Zr/Hf, Nb, Ta, and Li profiles, and a less evident, but still perceptible, jump at the G1a to G1b boundary, underlined, in particular, by Th or the Zr/Hf ratio. As a whole, the contrast was better seen when elements, though to be less sensitive to alteration (Th, Ti, Zr), were concerned. For example, Th varied from 0.8–1.9 ppm in G2–G3 to 10 ppm in the first meters of the G1 unit. Following a Th decrease upwards from 10 to 5 ppm in G1a, there was a jump to 9 ppm Th in the first meters of the Gb unit, followed upwards by a new decrease down to 6 ppm. Similar behavior was displayed by elements deemed weakly mobile in hydrothermal alteration, such as Nb and Ta, although less striking due, in particular, to the presence of some outliers, as also W and Sn (Figure 27). Based on the less mobile elements, the three sills of highly greisenised granites encountered by the SCB2 drill hole in the first 100 m above the granite roof (Figure 27) were very likely to be considered akin the G2 (or G3) unit.

Elements like As, Cu, or Zn showed little variations (other than erratic, with some outliers) in the lower section of the SCB2 profile, but displayed a distinctive enrichment in the upper part, in clear relationship with the correlative upward development of greisenisation (Figure 27).

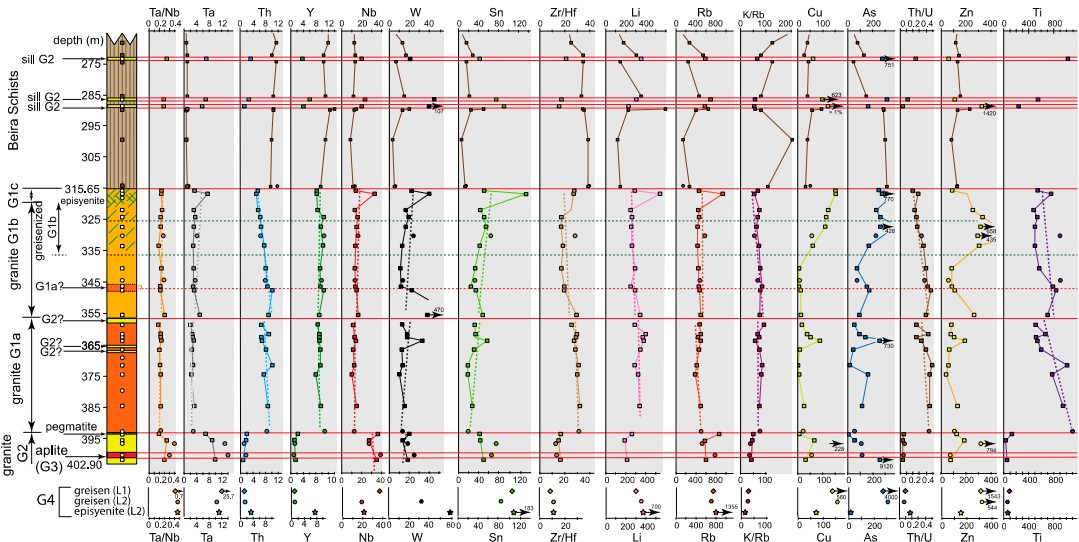

**Figure 27.** The SCB2 drill hole: trace element profiles. Data come from [16,17]; comparison with the G4 cupola composition: data from [18,56] (Table S5).

## 8.3. Rare Earth Elements

Only REE-analyses from the present study were used below. Concerning the SCB2 drill hole, the samples of our study were chosen as displaying the lesser alteration effects at the sample scale.

In contrast, the L2 level samples were all marked by a strong alteration. The data are given in Table S5, and the REE patterns are shown in Figure 28.

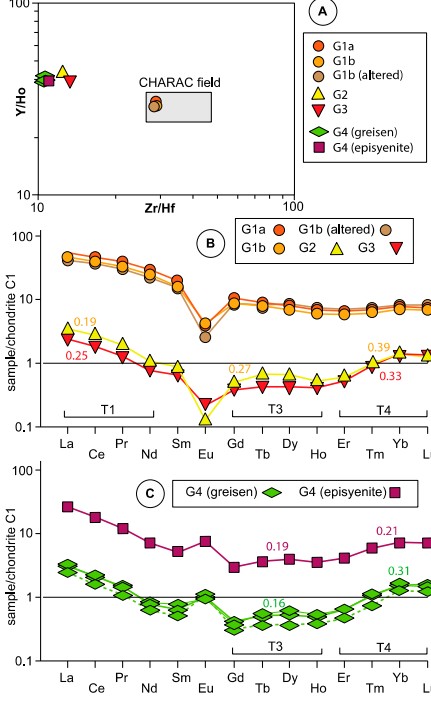

**Figure 28.** Patterns of RE variations (normalization to C1 chondrite [59]. Tetrad effect parameters Ti (i = I, II, III) are calculated following [60]; the effect is thought significant for Ti ≥ 0.2.

There was a clear-cut difference between the granites of the SCB2 drill hole, which were all characterized by a significant negative europium anomaly ($Eu_N/Eu^*_N$ around 0.3 for all samples) (Figure 28A), and the greisens or episyenite of the L2 level, which all displayed a strong positive europium anomaly ($Eu_N/Eu^*_N$ around 2.0 for all samples) (Figure 28B). On the other hand, there was also a clear opposition between the G1 suite, enriched in REE and, in particular, in LREE (average $\sum REE$ 59.5 and $La_N/Yb_N$ 6.17), relative to both the G2–G3 suite and the G4 greisens (average $\sum REE$ 3.45 and 3.05, and $La_N/Yb_N$ 2.09 and 2.03, respectively), with the episyenite, showing intermediate values ($\sum REE$ 26.25 and $La_N/Yb_N$ 3.57). Besides, as shown in Figure 28, a more or less significant tetrad effect was perceptible in the G2–G3 and above all in the G4 greisen (and episyenite) patterns, which could be linked to the non-CHARAC behavior of the same samples in a Y/Ho vs. Zr/Hf diagram [61] (Figure 28A).

## 9. Discussion

### 9.1. Quartz-Muscovite Relationships and the True Nature of the Greisen

#### 9.1.1. Greisen Definition

The name "greisen" is given to metasomatic alterations, affecting granitic rocks, and characterized by quartz-muscovite assemblages [62,63] and commonly attributed to the action of late magmatic fluids [2,64]. This genetic connection is currently integrated into the greisen definition. However, some works tend to demonstrate that metamorphic fluids (issued from the thermal aureole) may have been involved in the greisenisation process (e.g., [65,66]), despite the lack of robust evidence for the alleged magmatic fluid relationship. The quartz to muscovite ratio is variable in the greisens, and the name "quartz greisen" or "muscovite greisen" may be used when quartz or muscovite volume percent, respectively, is higher than 85% [63]. It has become increasingly clear that greisen is the equivalent in granites of the sericitic alteration in porphyry systems, and that consequently, it results from hydrolysis of the feldspars, i.e., of an $H^+$ metasomatism, ideally yielding a muscovite + 6 quartz assemblage (e.g., [67]). However, other processes are known to operate, including $K^+$ metasomatism in the field of muscovite stability, leading to muscovitisation (e.g., [67–71]), and they may occur in greisenised rocks (e.g., [72]). It is, thus, necessary, when dealing with quartz-muscovite assemblages, to ascertain the relative chronology of quartz and muscovite.

#### 9.1.2. Decoupling of Quartz and Muscovite in Panasqueira Greisens

There was good evidence for an intense silicification event in the G4 greisen. On the one hand, cathodoluminescence imagery showed that most of the quartz matrix in the greisen was not of magmatic origin (Figure 18). On the other hand, textural evidence showed that albite in the greisen was partially dissolved and replaced by quartz, without correlative crystallization of any mica (Figure 5). This quartzification event was equally perceptible in the granites of the SCB2 drill hole, with apparently a decreasing intensity downwards. Significant in the G1b unit, it was but faintly developed in the G1a unit, mainly expressed by digitations out of the quartz phenocrysts (and possibly the numerous quartz blebs in albite phenocrysts) and confirmed by cathodoluminescence images in which might be observed the same secondary quartz as in the best-developed greisens (Figure 18).

On the other hand, in the G4 greisens, consideration of the muscovite-quartz relationships led to the conclusion that quartzification was not coeval with muscovite crystallization, according to two series of observations that concerned (i) the muscovite frame, and (ii) the fluid inclusion (FI) petrography.

At the sample scale, all greisen samples were characterized (Figure 9D,E,G) by (i) the patchy distribution of muscovite aggregates and their ragged edges, with peripheral dissemination of small crystals, and (ii) their interconnection and tendency to organize along with definite directions. The muscovite fronts were in the same way lobated and ragged, and the muscovite aggregates of the greisen were merging with it (Figure 9E,G). Besides, as seen in Figure 9F, there was no apparent

compositional gap between the greisen and the muscovite. As muscovite resulted undoubtedly from quartz dissolution and replacement (episyenitization process: [69]), the same process produced the greisen muscovite. Owing to the identity of zoned muscovite (Ms1) in granites from the G1, G2, and G3 suite with the zoned muscovite from the G4 greisens, this conclusion must be extended to the whole granite pile at the top of the Panasqueira granite body.

On the other hand, the FI distribution in the G4 greisen was characterized by the systematic presence, in the greisen quartz, of patches of decrepitated FI (Figure 10A–C). These patches were separated continuously from nearby muscovite by areas of either clear quartz or quartz more or less loaded by non-decrepitated FI (the latter being equally observed, with variable abundance, in the decrepitated patches) (Figure 10). Overall, these observations pointed to the recrystallization of quartz at the muscovite contact and the late crystallization of muscovite, associated with quartz dissolution and recrystallization. Besides, they testified for an early timing of the quartzification event in the frame of the history of the Panasqueira hydrothermal system because it precedes the decrepitation event, which affects the quartz vein system after the main wolframite deposition event [42,73].

### 9.1.3. The G4 Cupola: A Pseudo-Greisen

In conclusion, the greisen in the G4 cupola was not a true greisen, in the sense that it did not result from the transformation of the granite feldspars into a muscovite-quartz assemblage in the course of a single process. It was indeed a quartzification in the first stage of the G4 alteration, and it was subsequently the subject of a K-metasomatism event in quartz undersaturated conditions, leading to quartz replacement by Li-muscovite, that is an episyenitisation process.

### 9.1.4. Lack of Relationships with Proximal Magmatic Fluids

Launay et al. [74] considered the quartz-muscovite assemblages in Panasqueira granites as a typical greisen assemblage, and, having recognized the unicity of the muscovite development throughout the granite pile, they judged that they were dealing with a classic "auto-greisenisation" process in relation with the magmatic-hydrothermal fluids issued from the main G1 body. The preceding observations were evidently at variance with this interpretation. What was more, it was evident, in the G1 granite suite, that a first alteration event preceded the incipient (pseudo)greisenisation. This first event was characterized by the development of a muscovite (Ms0)-carbonate-(chlorite) assemblage at the expense of the protolithionite, with the feldspars remaining stable. It was, therefore, not a greisenisation, being more akin to a classical propylitic event, attesting to the cooling of the granite beneath the solidus. It was thus excluded in any case that the (pseudo)greisenisation be related to magmatic-hydrothermal fluids issued from the G1 granite melt.

### 9.2. Subsolidus Reworking of Albite and Apatite

Besides quartz and muscovite development, there is other evidence for subsolidus processes in the granite units. The main observation relates to the composition of the feldspars. The nearly pure end-member compositions of both coexisting albite and K-feldspars are the clear indication of compositional re-equilibration at temperatures far lower than the solidus (<500 °C, [75]).

The zoning features of small apatite crystals included in albite from the G2-G4 suite might be equally be taken as evidence. These apatites were commonly zoned, with a corroded core, variably enriched in Mn and Fe, and a rim, close in composition to pure fluorapatite (Figure 20). It was also common to find, in the vicinity of these zoned crystals, small patches of zoned siderite. This zoning resulted from a subsolidus hydrothermal reworking of the magmatic apatite, probably coeval with a re-equilibration of the hosting albite. The occurrence of microfissural apatite related to the outer rims confirmed this interpretation.

In the same way, in the G1 granite, the albite aureole around the large apatite included in K-feldspar megacrysts might be taken as resulting from the reorganization of the perthite material. This conclusion was reinforced by the consideration of (i) the zoning of the apatite, with an external rim typically

resulting from the reworking of the pristine crystal, and (ii) the presence of overgrowths of siderite and APS (Figure 14). This interpretation might be extended to all the rims and the APS overgrowths of similar large complexly zoned apatite crystals in all the granite units. The existence of euhedral APS isolated in albite (Figure 14C) was further evidenced for the albite recrystallization. In the APS assemblages, the Al component (from surrounding albite?) and the LREE and U-Th (released from the alteration of monazite micro-inclusions?) might have been autochthonous. Still, Sr was introduced from outside, as was also the sulfur, thus equally testifying for hydrothermal reworking. A similar late introduction of Sr has been revealed in the Beauvoir RMG (French Massif Central) [76].

*9.3. Disentangling the Alteration Trends and Inferring the Magmatic Trend(s)*

Two ways were open to discriminate the magmatic trend(s) from the alteration effects. On the one hand, we could use those diagrams where the alteration minerals might be projected and their impacts subtracted, which are the so-called De La Roche diagrams (see Table S1). On the other hand, we could resort to tracers of magmatic fractionation, and in particular, to trace elements known to be the less sensitive to alteration effects, such as the thorium. These two approaches have been successively used in the following:

9.3.1. Major Elements Evidence

Granites of the G1 suite were severely affected by an early alteration, impacting the protolithionite, whereas other granites (G2 to G4) were spared by it. Reciprocally, the latter were more affected by the (pseudo)greisenisation process. These considerations provided the basis to explain the different trends displayed by the Panasqueira granites in the diagrams of Figure 25. In these diagrams, it was indeed possible to draw "alteration vectors" and to check how they apply to the measured compositions. As seen in Figure 25, the distribution of the data plots was reasonably explained by the combination of alteration vectors. The succession of processes was: (i) the protolithionite alteration (for the G1 suite), (ii) the silicification followed by (ii) muscovitisation (for all the suite). When considering the initial compositions that were either preserved (or at least, weakly disturbed) from alteration (a few G1 analyses) or reconstructed by inverting the alteration processes (G2-G4 sequence), the magmatic trend for the Panasqueira granites could be defined. Figure 25 shows that depending on the diagram, different alteration vectors might be better expressed than others. For example, in the AB diagram (Figure 25A), as the effects of quartzification being directly opposed to that of albite dissolution, they canceled each other, whereas, to the contrary, they were additive in the QP diagram (Figure 25B). In the Q*3B*3F*3 diagram in Figure 25C, the effects of albite replacement by muscovite Ms1 in granites of the G1 suite were particularly sensitive. In contrast, they remained practically invisible in the AB diagram in Figure 25A. The effects of dequartzification and correlative muscovitisation were additive in all diagrams, and this was at the origin of the pseudo-greisen trends.

The interpretation in the three representations was always consistent with the petrographic observation. In the G1 suite, the early muscovitisation (protolithionite alteration) was the dominant process, whereas, in the G2-G4 series, the later quartzification-dequartzification (muscovite Ms1) sequence was by far the main process. The quartzification was still significant in G4, and muscovitisation was more intense in G2 than G4. In the same way, the fact that the same magmatic trend obtained in the three different diagrams in Figure 25A–C also gave more confidence in the reliability of the reconstruction. The resulting magmatic trend was characterized by increasing albite content and decreasing quartz content (Figure 25B), similar to the RMG trends of the Beauvoir type, also characterized by an increasing fluorine content in the melt (reference trend plotted in Figure 25A–C). The most evolved unit of the Panasqueira suite was the G4 unit (now, transformed into the greisen cupola), which seemed to be close to the Argemela RMG (Figure 25).

9.3.2. Trace Elements Evidence

Thorium, one of the least mobile elements in hydrothermal processes, is expected to decrease with fractionation (mainly of monazite) in highly peraluminous low-temperature magmas [77] and references therein. Considering the magmatic trend reconstructed in Figure 25, Th would be a reliable marker of the fractionation process, to which we referred the behavior of other trace elements. In the same way, as far as fractionation in the peraluminous RMG suites was controlled, among others, by the continuous evolution of Li-bearing micas from tri- to di-octahedral compositions, a parameter like B*3 (in Figure 25C), which precisely expressed the balance between trioctahedral (when positive) and dioctahedral (when negative) micas, would be equally well-fitted suited to mark the Panasqueira fractionation process. At the same time, it had the potential to track hydrothermal alterations involving muscovite. The plot of Th vs. B*3, as shown in Figure 29, confirmed this interpretation.

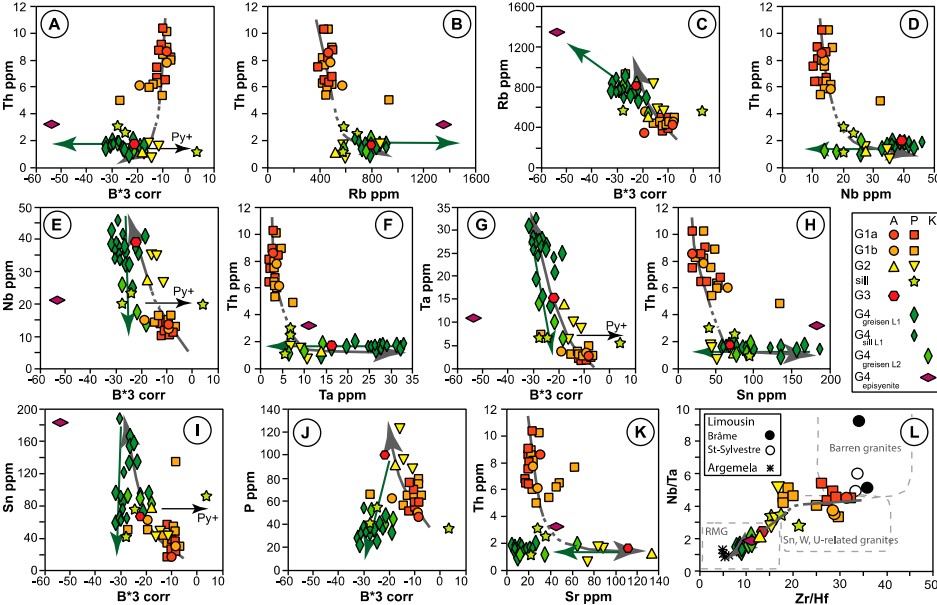

**Figure 29.** Disentangling magmatic and alteration trends in the Panasqueira suite using trace elements (Table S4). Trace element variations are potential markers of the magmatic fractionation: inert element Th, B*3 parameter (from the Q*3B*3F*3 diagram in Figure 25C), and Nb/Ta and Zr/Hf ratios. Estimated trends: grey arrow, magmatic fractionation trend, green arrow, a hydrothermal trend in the greisenised facies. (**A**) Th vs. B*3 diagram, demonstrating that it is effectively possible to disentangle magmatic and alteration trends using these parameters. (**B,C**) Th vs. Rb (B) and Rb vs. B*3 (C) diagrams, showing that greisens are characterized by an increase in Rb contents beyond the values reached in the fractionation process. (**D,E**) Th vs. Nb (D) and Nb vs. B*3 (E) diagrams, demonstrating that greisens are characterized by decreasing contents from the high values reached through the fractionation process. (**F,G**) Th vs. Ta (F) and Ta vs. B*3 (G) diagrams, showing that Ta behaved as Nb in both fractionation and greisenisation processes. (**H,I**) Th vs. Sn (H) and Sn vs. B*3 (I) diagrams, showing the same evolutions as Nb and Ta. (**J**) P vs. B*3 diagram, showing that P (as apatite, see Figure 24), although effectively concentrated throughout the fractionation process, has very efficiently leached out the more evolved facies by the alteration process. (**K**) Th vs. Sr diagram, showing that Sr (being mainly sequestered into apatite) behaves just as P. (**L**) Nb/Ta vs. Zr/Hf diagram, very efficiently displaying the full fractionation sequence of the Panasqueira suite (G1a-G1b-G2-G3-G4). Comparison with reference granites shows that G4 is a true RMG (very similar to the nearby Argemela RMG).

Binary diagrams from Figure 29B–K corroborated the existence of a single magmatic trend from G1 to G3 and which might be extrapolated to G4 using the values in the greisenised facies of the L2 level. This trend was typically composed of two segments, with a gap and a strong inflection, between the G1a-G1b suite on the one hand, and a G2-G4 suite, on the other, both monitored by

either Th or B*3 decrease. An increase in Rb (Figure 29B,C), and in particular, Nb (Figure 29D,E), Ta (Figure 29E,F), and Sn (Figure 29H,I), was significant. Extrapolation to G4 compositions showed that Ta and Sn concentrations were particularly high in this facies. An increase of the phosphorus content was observed from G1 to G3 (Figure 29J), paralleled by a strontium increase (Figure 29K). The binary diagrams in Figure 29 equally displayed evidence for a strong hydrothermal reworking in relation with muscovitisation (Ms1), particularly evident in the greisenised G4 unit, as could have been expected, but also visible in the G2 unit (in particular, in the three sills of the upper part of the SCB2 drill hole). Except for Rb, which was typically enriched in the hydrothermal process (Figure 29C,D), muscovitisation generally resulted in intense leaching of the elements, previously enriched during the magmatic evolution (Figure 29E–K). This leaching was, however, of similar intensity for Nb and Ta (Figure 29E–H), with the effect that the Nb/Ta ratio was practically not modified. It resulted that the Panasqueira magmatic trend was well-defined in the Nb/Ta vs. Zr/Hf diagram (Figure 29L), where the convergence of the G4 unit with the Argemela RMG was once again remarkable.

### 9.3.3. Conclusions

The granite geochemistry at the top of the Panasqueira granite body, such as imaged by Ribeiro [49] (Figure 2), was representative of an RMG fractionation trend of a peraluminous high-phosphorus (PHP) granite type [3]. Such a conclusion was consistent with the phosphorous enrichment correlated to the increasing fractionation (Figure 29I) and the abundance of fluorapatite. Besides, the Mn contents were also compatible with the increase in the degree of fractionation (Figure 20). This trend ended with the highly fractionated G4 cupola, e.g., the Barroca Grande granite of [14], which might also be compared to the Argemela cupola (Figures 25 and 29L). It was also corroborated by (i) the presence of P-rich feldspars in the more fractionated terms [78] and (ii) the presence in large magmatic apatite crystals of micro-inclusions of the alluaudite and wagnerite group minerals, found in pegmatites of the LCT-type, i.e., equivalent to the PHP-RMG [3].

It seemed likely that the Panasqueira suite evolution was driven by a fractionation process involving quartz, feldspars, and micas, as in other RMG suites [3,79]. The curvature in the trends observed in Figure 25, and especially in Figure 27, was evidence for a change in the fractionating package between the G1 suite and the G2 to G4 sequence, most likely due to the relay of protolithionite by a Li–Fe-muscovite in the magma, as indicated by the good monitoring of the fractionation process by the B*3 parameter. This inferred Li–Fe magmatic muscovite must be distinguished from the late pervasive Li–Fe hydrothermal muscovite. It is a fact that in the Argemela RMG, which is so similar to the Panasqueira G4, the main mica is a zoned Li-muscovite [55,80]. Support for this interpretation came from (i) the apparent lack of protolithionite in the more evolved facies, (ii) the observation of (rare) Fe–Li-muscovite relics in the snowball quartz in G4, and (iii) the existence in some muscovite Ms1 crystals from the greisens of Li–Fe-cores (e.g., Figure 6I). Such features could explain why the G2 to G4 suite was free of the early muscovitisation process (Ms0), which only affected protolithionite. There was, besides, a clear gap in the fractionation trend in most of the trace element binary diagrams in Figure 29, which likely recorded a time gap between the end of the main granite emplacement (G1 suite) and the subsequent intrusion of the G2 to G4 suite. In turn, this could mean that there was indeed a renewed sourcing of rare metal melts in the system due to fractionation or other mechanisms involving the source [81].

### 9.4. Origin of the Quartz Cap

The quartz greisenisation event might be a clue to the interpretation of the enigmatic "quartz cap" described at the top of the greisen cupola [11], the only workers to have seen it. The latter described it as massive white quartz clouded with muscovite and affected by subhorizontal sheeting with "muscovite metasomatism". This quartz cap could well be the result of quartzification. Indeed, [11] (p. 1741) mentioned that there was a continuity between the quartz cap and some quartz veins of the main system. But it was more likely that these "veins" were indeed quartzified sills issued from the cupola,

emphasizing then its initial magmatic nature. Such a conclusion was further suggested by another observation [11] that "At some spots along the accessible edges of the cap, large euhedral quartz crystals, some more than 20 cm long, can be seen along with the hanging-wall contact. These appear to have perched on the schist and grown inward toward the cap interior. Such crystals are most easily seen where later quartz filling in around them, is rich in fine-grained muscovite" (p. 1741) and that "At one location, these big crystals continue out a short distance as a lining of the hanging wall of a thick vein that joins the cap" (p. 1741) (Figure 3A). The cap was considered to be of hydrothermal origin, and these giant quartz crystals were to represent the earlier crystals of this quartz infilling [11]. It seemed, however, clear that, to the contrary, these observations reinforced this interpretation of the quartz cap. These giant quartz crystals would have been the only preserved remnants of a now destroyed stockscheider. In this connection, [11] underlined that the "quartz cap", a thin body (14 m thick) with minimal development (53 × 73 m in the plan), had "a flat floor and an arching roof", and that the contact with the underlying greisen was sharp (Figure 3A). All these data pointed to an interpretation of the quartz cap as the testimony of a past stockscheider at the top of the cupola.

### 9.5. Timing of Granite Emplacement

#### 9.5.1. Relative Chronology and Duration

The extended thermal aureole at Panasqueira is likely interpreted as the consequence of the intrusion of the granite body modeled by gravimetry [34]. It is currently observed, in the tourmalinized wall-rocks of the Panasqueira vein system, that cordierite spots of this aureole are overprinted by tourmaline [40–45]. Consequently, the main granite body (therefore, the G1 suite) must have been emplaced earlier than the tourmalinization event. On the other hand, the granite is not deformed, and, therefore, later than the D3 event, responsible for the large upright folds at the regional scale. It could nevertheless have been late-kinematic, but in any case, the G4 cupola is later than the late-kinematic seixo-bravo system, which is present as enclaves at the L2 level, and cross-cut by aplites at the L1 level [15].

The G1a and G1b members of the G1 suite presenting, in Figure 29, the same range of Th contents, and thus having the same degree of fractionation, might, therefore, represent coeval intrusive batches. As discussed above, the G2 and G3 units might be interpreted as more fractionated members of the same suite and, thus, probably set up later than the G1 suite. Observation of the geochemical profiles along the SCB2 drill hole showed that, for elements like Ta, Th, Y, or Ti, there was at the G2 to G1a boundary a significant shift in G2 compositions towards values more typical of G1a (Figure 27). Limited mixing processes at the very edge readily explained this behavior: it was, thus, consistent with a later emplacement of the G2 unit, although obviously at a time when the G1a unit was still not wholly crystallized. The most evolved facies of the suite, the Barroca Grande (G4) granite cupola, was likely the latest emplaced, in agreement with its apical position.

In the geochemical profiles in Figure 27, a series of elements displayed distinct trends very suggestive of fractionation trends in either the G1a or the G1b unit (e.g., Th, Ti, Zr/Hf, decreasing upwards, or Nb, W, Sn, increasing upwards). These trends might represent the successive emplacement of more and more evolved fractionated batches issued from the differentiation of a single melt at greater depth (most likely by a combination of Bagnold and filter-press effects in feeder dikes: [80]). The Nb/Ta vs. Zr/Hf diagram in Figure 29L showed that, in the G4 unit, the facies of the L2 level was slightly less evolved than at the L1 level above, which might be interpreted in the same way. The sheet-like morphology of the granite units was in good agreement with this view, which was equally in line with the worldwide growing evidence that many granite plutons were indeed built by accretion of successive rather thin sills, ≤100 m thick, emplaced through feeder dikes (e.g., [82–87]).

In the course of such an accreting process, the time interval between successive intrusion pulses is currently estimated between 10 and 60 kyr [84,86–91]. If such a mechanism was active for the entire emplacement of the Panasqueira pluton, the total duration of the event could not have been longer

than ~1 Myr (comprised between 200 kyr and 1.2 Myr, assuming sills of about 100 m thick), and the emplacement of the G1 to G4 sequence would have been even shorter.

### 9.5.2. Absolute Ages

Until recently, the age database at Panasqueira has been somewhat limited, consisting of a loose K–Ar age on the Panasqueira Granite of 290 ± 10 Ma [92]. The Argemela age could place a constraint, but it is not known; the microgranite is only suspected of having been emplaced in the 305–300 Ma range [31], as also supported by the 303 ± 6 Ma K–Ar muscovite age of [93].

Recent dating at 305.2 ± 5.7 Ma of the earliest hydrothermal event (tourmalinization) in the Panasqueira vein system (U/Pb on rutile: [45]) constrains the emplacement of the main granite body (of which G1 units are the upper part) to be earlier since minerals from the thermal aureole are overprinted by tourmaline. Owing, on the other hand, to the relationships with regional deformation, it cannot, however, be older than 310–305 Ma and could have been more or less coeval with the nearby batholiths of Sierra de Estrela, Fundao, or Castelo Branco. The granitic rocks from Castelo Branco pluton are 310 ± 1 Ma old, obtained by U–Pb (ID-TIMS) in separated zircon and monazite crystals [94]. The age of 296.3 ± 4.2 Ma obtained (U/Pb) on apatite from the G1 granite [15] is thus surprising. It is interpreted as dating the emplacement of the granite considered as a single intrusion. It is also similar to a 294.5 ± 5.3 Ma age (U/Pb) obtained on apatite from the mineralized veins. It is, therefore, considered as validating the concept of a magmatic/hydrothermal system at Panasqueira [15,41,42], but is, however, contradictory to the above-cited evidence.

There is no specification [15] of the kind of apatite crystals used for G1 dating, except that they comprise "magmatic apatite" from the 2-mica porphyritic granite (=G1a) and "altered magmatic apatite" of "the greisen" (presumably, the uppermost G1b facies from the present study, see Figure 27). All the alterations affecting the granite are thought [15] to have been the result of an "auto-greisenisation process", involving magmatic fluids. However, as we have shown above, the pristine magmatic apatite experienced a strong subsolidus reworking, and therefore, the obtained U/Pb age [15] most probably recorded this alteration event. The previous K/Ar age of ca. 290 Ma obtained on G1 [92] must also be interpreted as recording this alteration event.

The age at ca. 294 Ma [15] was also very similar to the 294 ± 1.2 Ma age [44,95] for the main muscovite development in the Panasqueira vein system. It seemed, therefore, likely that muscovite in the vein system and muscovitisation in the granite system were two manifestations of the same hydrothermal event at ca. 294 Ma.

### 9.6. Granite Potential as a Source of Metals for the Panasqueira Mineral System

The timing of the granite emplacement (see above) precluded a direct role of magmatic-hydrothermal fluids issued from the different Panasqueira intrusions. What was more, the record of magmatic fluid-rock interaction, at the outer contact of the G4 granite with the surrounding rocks, was not in favor of a massive export of rare metals at the time of G4 emplacement. In fact, to the exception of a rim of zinnwaldite metasomatism, there was no mineral evidence for any output from the nearby magma: the absence of any rare metal-bearing mineral was striking. In the same way, as seen in the geochemical profiles of the SCB2 drill hole (Figure 27), there was no change in the composition of the overlying schists pile at the very contact with the granite roof, to the exception of a clear Rb enrichment. It was only in the close vicinity of the thin G2 sills above that a limited effect might be observed at the very contacts, such as enrichments in W, Sn, Li, Rb, but not in Nb or Ta. Still, it might as well be a mere reflection of the intense alteration affecting these sills, as shown by concomitant enrichments in Cu, Zn, and As (Figure 27).

Hydrothermal fluid-rock interaction at high temperature had the potential of releasing rare metals from Panasqueira granites, either the early muscovitisation (Ms0) process in the G1 suite or the quartzification-muscovitisation process in the G4 RMG. Muscovitisation of the G1 protolithionite was associated with the trapping of released titanium as rutile crystals, with Nb enrichments in the pyramid

faces, but also Ta and W in lesser proportion. Rather than proceeding from the incoming hydrothermal fluid, these rare metals were likely to have equally been released from the destroyed protolithionite. In granite melts, niobium behaves as a compatible element, being preferentially partitioned into biotite [96]. Besides, as the habit of the newly formed rutile is dominantly prismatic, it results that the bulk content of rare metals in these crystals has remained limited [44]. As a consequence, the fluids escaping from the altered granite were probably enriched in rare metals, in the first place, W and Ta, which were less trapped in rutile than niobium and were, therefore, able to contribute to the endowment of the Panasqueira system.

The G4 unit was an RMG, very similar to the nearby Argemela Granite, and, consequently, was enriched in rare metals, and in particular, Nb and Ta. Thus, the G4 granite contained probably rare metal minerals (columbite-tantalite), as the G2-G3 ones in the Panasqueira system, or in the Argemela RMG, in which the rare metal minerals were hosted by magmatic apatite or albite or a Li-mica [32]. Yet, rare metal minerals were typically lacking in the altered G4 cupola, and there was clear trace element evidence for rare metals having been thoroughly leached through the alteration process. This leaching was essentially due to the quartzification process. Indeed, it might be suspected that late muscovitisation was able to partly compensate for the leaching by adding a rare metal content, thus minimizing the earlier alteration effects. The Sn (Figure 29H,I) and W enrichment in the episyenite were thus corroborated by the local presence of wolframite or cassiterite associated with the episyenitization.

It follows that when they were flowing out from the G4 cupola, the fluids responsible for quartzification could well have extracted rare metals, in particular niobium, owing to the high Nb to Ta ratio in the Panasqueira columbite-tantalite (see above). A contribution of these rare metals to the Panasqueira system could, therefore, be envisaged, provided that the cupola quartzification was coeval with the mineralization process. As for the G1 case, two facts were in favor of the involvement of G4. First, the quarzification affecting the mineralized vein selvages is an early process [43], and this quartzification may tentatively be ascribed to the same process as in the G4 granite. Second, the study of early rutile found in tourmalinized vein selvages has demonstrated a periodic input of an Nb-rich fluid, differing from the main hydrothermal fluid responsible for the alteration [44], and it is appealing to relate this Nb-rich fluid to the alteration process in G4.

In summary, it might be assumed that the Panasqueira granites had the potential to play a passive role for the mineralizing system, working as metal sources through HT hydrothermal fluid-rock interaction. Whereas the minimal volume of the G4 cupola precluded a significant role for it for the leaching of W, the larger G1 body could well have provided a substantial part of the Panasqueira rare metal endowment. With an average 13 ppm W and a volume of 32 km$^3$, it represented, in particular, a total of ~1Mt W, more than was needed to feed both the quartz-wolframite system (120–150 kt W) and the anomalous halo of at least 350 kt W described by [7]. However, such a conclusion is somewhat optimistic because (i) it is unknown if all the body is as rich in W as is its apical part, (ii) the released quantity would depend upon both the volume involved in the alteration process and the efficiency of the leaching and metal deposition—three parameters that remain unknown. Owing to the remnant 13 ppm W in G1, it might be expected that only a few more ppm could have been leached in the upper part accessible to our observations. If it was supposed that only a more realistic 500 m thick layer of the G1 body released only 2 to 5 ppm W, the delivery to the hydrothermal system would have been no more than 75–190 kt W. This mass was just sufficient to provide the tungsten content in the veins (supposing an unrealistic 100% efficiency of trapping nevertheless) but unable to feed the hydrothermal halo. It must, therefore, be concluded that the G1 granite could at best have been a significant, but not unique, contributor of tungsten to the Panasqueira deposit. This conclusion found similarities with the possible contribution of Sn-rich late magmatic titanite from the Qitianling granite to the genesis of the world-class Furong Sn deposit (SE China) when hydrothermally altered at subsolidus (ca. 400 °C) temperatures [97].

**Supplementary Materials:** The following are available online at http://www.mdpi.com/2075-163X/10/6/562/s1, Table S1: Chemical composition of micas; Table S2: Chemical composition of apatite, alumina-phosphate-sulfate,

and monazite; Table S3: Chemical composition of rutile and columbite-tantalite; Table S4: Major element composition of granites and greisens; Table S5: Trace element composition of granites and greisens; File S1: Explanation for the use of the de La Roche chemical-mineralogical class of diagrams; File S2: Analytical methods. References for Files S1 and S2: [97–101].

**Author Contributions:** Conceptualization, C.M., M.C. (Michel Cuney), M.C. (Michel Cathelineau); methodology, A.L.; investigation, data acquisition: all authors; writing—original draft preparation, C.M., M.C. (Michel Cuney). All authors have read and agreed to the published version of the manuscript.

**Funding:** This work was funded by the ERAMIN project NewOres funded by ANR (ANR-14-EMIN-0001), and Labex Ressources 21 (supported by the French National Research Agency through the national program "Investissements d'avenir") with reference ANR – 10 – LABX 21 —LABEX RESSOURCES 21. The MicroXRF used is a piece of equipment co-funded by ICEEL (Carnot institute)-CREGU-Labex Resources 21, Lorraine Région, and FEDER.

**Acknowledgments:** Three anonymous reviewers are warmly thanked for their careful reading and constructive recommendations. We are most grateful to Beralt Tin and Wolfram S.A. for permitting access to the Panasqueira underground mine.

**Conflicts of Interest:** The authors declare no conflict of interest. The funders had no role in the design of the study; in the collection, analyses, or interpretation of data; in the writing of the manuscript, or in the decision to publish the results.

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
