# Peer review of "The Panasqueira Rare Metal Granite Suites and Their Involvement in the Genesis of the World-Class Panasqueira W–Sn–Cu Vein Deposit: A Petrographic, Mineralogical, and Geochemical Study"

_minerals, doi:10.3390/min10060562_

Round 1

Reviewer 1 Report

General remarks on the manuscript "The Panasqueira Rare Metal Granite Suites and their Involvement in the Genesis of the World-Class Panasqueira W-Sn-Cu Vein Deposit: A Petrographic, Mineralogical and Geochemical Study by Marignac et al. for publication in Elements

The investigation of Marignac et al. provides a thorough and explicit petrographic and geochemical description of a granitic suite, which experienced a distinct secondary alteration possibly leading to the generation of a hydrothermal rare metal deposit. The study supplies a detailed overview on the different alteration structures and textures, which is supported by numerous microprobe analyses. Geochemical analyses of main and trace elements highlight the primary and secondary features of the rock suite and numerous geochemical trend and discrimination diagrams unravel the magmatic and post-magmatic evolution of the rocks. The observations are supported well by the large amount of data and the vey careful conclusions are quite comprehensible. The study is performed in a professional way but the scientific English language of the paper should be improved by a geo-scientist of English mother tongue.

Despite of these positive aspects, the study suffers from several, scientific, formal and stylistic shortcomings, which necessitate major revisions. Enrichment of rare elements in granitic melt does not always have to originate from fractionation processes but can also be due to higher magmatic melting temperatures, primary enrichment of the rare elements in a chemically altered protolith of the granite or changing water activity during melting. This should be discussed in this study and I have made a proposition below (see Specific points, Page 39, lines 1145-1149).

The manuscript is very long and especially the petrographic descriptions could be shortened to a certain extent. Part of these descriptions could – for example – be placed in the appendix, which also holds true for some figures and diagrams. The main point of criticism here concerns several figures, which lack scale bars, captions and even microphotographs! Furthermore, there is a description of a Figure (pages 30 and 31, lines 894-902), which cannot be found in the manuscript nor in its appendix. Also, several formal and grammatical errors have to be improved (see below). On the one hand side, the reviewer really appreciates the exhaustive petrographic diagrams and the nicely done assignment to the different alteration processes. On the other side, these observations have a great potential to unravel the geochemical character of these processes in detail, which was hardly accomplished in this study. Mineral zonation patterns like the congenial zoning of white mica, apatite or zircon are petrographically well described but the link to the mineral analyses and the geochemical processes generating these patterns is only partially performed. For these presented mineral alterations and zonation patterns, geochemical reaction could even be formulated (which, however, would be beyond the scope of this paper).

Absolutely necessary is the re-writing of the Abstract, which does not at all describe the findings and the results of this study. It should be stated, that the investigated granite consists of several layers (probably magmatic pulses), which can de discriminated geochemically and which suffered distinctly different, polyphase alterations. These findings totally lack in the abstract, where only the granite types G1 und G4 are mentioned. After my opinion, the main compelling point of this study is the fact that – despite of the extreme late- and post-magmatic overprint – primary magmatic as well as secondary alteration trends can unambiguously be discriminated by the geochemistry of main and trace elements! Quote this in the Abstract, please!

Due to these points I consider to accept the manuscript given that major revisions are performed. If that is the case, the investigation will be a valid contribution to the topic of secondary alteration and ore forming processes in granites.   

Specific points to be improved in the manuscript

Whole text: The authors often use the expression "sub-euhedral" or "subeuhedral", which is a bit uncommon. I suppose that the term "subhedral" should be used. For a consistent nomenclature I would also propose to use "biotite" instead of "black mica".

Page 1, lines 23 and 24: The expression "terms" and "term" for a granite types seems a little weird; they should be replaced by the expression "rock type".

Page 2, line 48: Replace "done" by "performed".

Page 2, line 53: Replace "ere" by "were".

Page 3, lines 87-88: Rephrase the sentence to: "At 10 km to the east, the small cupola of Argemela, a rare metal granite (RMG) associated with tin mineralization and minor wolframite veins, is located [32-33]."

Page 4, line 93: Rephrase the sentence to: "Petrography of the upper granite body according to [11-12]."

Page 4, line 97: The description of Figure 2-C (SCB2 drill hole) is lacking in the captions!

Page 4, line 109: Replace "crosscut" by "crosscuts".

Page 5, line 119: Replace "made" by "consisting". 

Page 5, lines 135-138: The Captions of Figure 3 are wrong: Figure 3-A shows the quartz cap and Figure 3-B shows the greisenized granite with the mica schist layers! Please correct the Captions.  

Page 6, line 155: Replace "may" by "can".

Page 7, line 220-231: In the captions of Figure 4, all abbreviations should be explained! "Znw" is first explained in the captions of Figure 8 and "Po" (Figure 4-A) is not explained at all. Probably, "Po" means "protolithionite", for which the abbreviation "Prl" is used in the captions. Please correct that.

Page 7, line 232: Replace "rock" by greisenized granite" because there are different rocks shown in Fig. 4-A.

Page 8, line 245: In Figure 5-A, there is no abbreviation "Qtz" as noted in the captions; please correct that.

Page 8, Figures 5-B, 5-C, 5-D, 5-E, 5-F and 5-G: Please add mineral abbreviations for topaz, albite and quartz in the microphotographs.

Page 9, line 254: Replace " Z3succession" by " Z3 succession".

Page 9, Figures 6-D, 6-E, 6-F, 6-G, 6-H, 6-I, 6-J and 6-L: Please add scale bars.

Page 9, Figure 6-H: No microphotograph is recognizable!

Page 9, Figure 6-L: Please add the letter "L" in the figure.

Page 10, line 276: Replace "display" by "displays".

Page 12, line 334: Replace "strongly" by "distinctly".

Page 12, Figures 9-A, 9-C, 9-D, 9-E, 9-F and 9-G: Please add scale bars.

Page 12, line 359: Add "(Wfm)" behind "wolframite".

Page 13, line 362: Delete the comma behind "lace-shaped".

Page 13, line 367-368: Rephrase the sentence to "Note the intimate association of the muscovite with the wolframite crystals from a wolframite pseudo-vein,…"

Page 13, line 376: In this line, the term "relics" is used whereas you find the term "relicts" in the text before. I suggest that throughout the text, the expression "relicts" should be used consistently.

Page 13, line 387-389: This short summary on the two G1 granite facies (G1a and G1b) must be expanded! As shown in the following the authors describe the very important third facies G1c, i.e. the episyenite (lines 519-555 of the manuscript). Despite of its small extension, the G1c facies is nevertheless very important and should also be mentioned here!

Page 14, line 393: The term "expansion" is incorrect here because it is an epitaxial overgrowth.

Page 16, line 438: Delete "they".

Page 16, Figure 13: Mark rutile (Rt) in the BSE images and characterize the surrounding material in Figure 13-A (Qtz?).

Page 17, Figure 14: Please mark all minerals in the BSE images with mineral abbreviations.

Page 19, lines 521-532: Rephrase the sentence to " The latter is fine-grained, with a quartz-feldspar (0.2-0.3 mm) -muscovite matrix, in which albite microphenocrysts (0.5-1 mm), larger phenocrysts of albite (≤ 5 mm) and magmatic quartz (1-3 mm) are dispersed."

Page 19, lines 526: Replace "aureole" by "aureoles".

Page 19, lines 532: Replace "aureole" by "aureoles".

Page 19, lines 534: Replace "poecilitically" by "poicilitically ".

Page 19, lines 557: Replace " fine-to" by " fine- to ".

Page 19, lines 558-559: Move "is embedded" at the end of the sentence.

Page 20, line 572: Delete "however,".

Page 20, line 578: Replace "numerous prismatic cores" by "prismatic dark core sections".

Page 20, line 580: Replace "emplacement" by "position".

Page 21, line 600: Replace "corroding" by "corroded".

Page 21, line 606: What is "the latter"? According to the previous sentence it would be albite but a Mg-rich albite is very uncommon!

Page 23: In Figures 18-A, 18-C, 18-E, 18-F, 18-G, 18-H and 18-I the cathodoluminescence images are missing! In the upper part of Figure 18-B there is a white square.

Page 27, line 799: Replace "whereas" by "where".

Page 28, line 809: Rephrase the sentence to: "Large rutile crystals occur in the G1 suite where they are the only bearers of rare metals."

Page 29, line 862-863: Rephrase the sentence to "In Figure 26 significant differences in the trace element contents are evident between the SCB2 granites and the cupola greisens,…"

Page 29, line 867: Replace "contrasted" by "evident".

Page 29, line 870-872: Please rephrase and split up the sentence: "For example, Th changes from 0.8-1.9 ppm in G2-G3 to 10 ppm in the first meters of the G1 unit, and, while in G1a, Th increases upwards from 10 to 5 ppm, there is a jump to 9 ppm Th in the first meters of the Gb unit, followed upwards by a decrease down to 6 ppm." It is hard to understand!

Pages 30 and 31, lines 894-902: I do not understand the relation of this paragraph to the rest of the geochemistry before and afterwards! Some strange "AB diagram" is referred to, which I could not find in the manuscript. The hint on "Appendix A1 for explanations" (line 895) is still more confusing as Appendix A only contains mineral analyses (i.e. the chemical composition of apatite, alumino-phosphate-sulphate and monazite) and no diagrams! Either anything was forgotten or this paragraph is a relic from an earlier version of this manuscript.

Page 33, line 927: Rephrase the sentence to "Only REE-analyses from the present study were used below."

Page 35, line 959: Replace "yielding" by "leading".

Page 37, lines 1055-1057: The meaning of the sentence " In interpreting the diagrams in Figure 25, it must be realized that, depending on the diagram, some alteration vectors may be best expressed than others." is not quite clear. The chapter below describes opposing trends of alteration vectors, which may even be wiped out by different alteration processes. Therefore, the sentence above should be rephrased expressing that an interpretation of the different vectors is distinctly aggravated by the polyphase alteration processes.

Page 37, line 1085: Replace "confirm" by "confirms".

Page 38, line 1101: Replace "behave" by "behaves".

Page 39, lines 1145-1149: The authors write: "There is, in addition, a clear gap in the fractionation trend in most of the trace element binary diagrams in Figure 29, which likely records a time gap between the end of the main granite emplacement (G1 suite) and the subsequent intrusion of the G2 to G4 suite. In turn, this could mean that there was indeed a renewed sourcing of rare metal melts in the system." This clear gap is clearly evident and very important. However, it does not necessarily have to result from fractionation but may also be due to other mechanisms, i.e. a higher temperature for the initial magmatic melting of the subsequent granite intrusion (which mobilizes Sn from minerals like biotite, titanite or rutile), or the primary enrichment of Sn and W in a chemically altered protolith, or a changing activity of water during melt reactions. Those mechanisms were investigated in detail in a recent study of Wolf et. al. (Wolf, M.; Romer, R.L.; Franz, L. & López-Moro, F.J. (2018): Tin in granitic melts: the role of melting temperature and protolith composition. Lithos, 310-311, 20-30.), which should be cited here!

Page 39, line 1166: Replace "quartz" by "quartz crystals".

Page 40, line 1206: Put "." behind the sentence.

Author Response

We have carefully taken into account most of your suggestions, we warmly thank you for attentive reading, and pertinent remarks and suggestion. There were however a few exceptions, that we explain below.

Concerning the length of the manuscript, we have put the analytical methods as an Appendix 6 and we have tried to reduce a little the text. However we are convinced that we have to keep all the figures with the detailed mineralogical descriptions (which have been highly appreciated by all the reviewers), otherwise it will be very tidy for the reader to open each times the annex to find the relevant figure corresponding to the petrographic detail he would like to verify, and they are also important as we are using diagrams which are not only geochemical but mineralogical-chemical. For a good understanding of the trends defined in these diagrams the mineralogical observations are also essential. We think that such detailed mineralogical investigation combined with mineral-chemical diagrams is rare in the literature and thus should be preserved as such !

The abstract has been extended.

The scale bars have been added to the figures.

Quality of the microphotographs: you have had some difficulties to examine the microphotographs, and especially those from Figure 18 with the images of cathodoluminescence, in the pdf version of the manuscript. The resolution of the pictures is very bad in the pdf and explain your remarks! We hope that this problem will be alleviated when publishing.

Comment on line 393. We disagree with the interpretation of the large albite fields around apatite as “epitaxial overgrowth”: it looks to us as the result of subsolidus reworking of the epitaxial perthitic array.

Comment on lines 1145-1149. You are right to insist on the importance of processes acting in the magma source. This is however out from the scope of our work, and we only add a few notations (and the indicated reference) to our previous statements.

Reviewer 2 Report

This is an excellent analysis of the granites and greisen in the Panasqueira W-Sn-Cu deposit system. I have made minor editorial notes in the pdf that the authors should consider. Citations using #'s should read better that the authors have presented. The rest are suggestions and technical edits to consider only. I do think that the discussion could include opinions (based on their evidence) for fluid fractionation versus extreme (igneous or crystal) fractionation processes. Analysis of Monecke's more recent paper on Tetrad effects and Boulliard's Nb-Ta arguments need more scrutiny.

Author Response

We have carefully taken into account most of your suggestions, we warmly thank for your attentive reading, and the pertinent remarks and suggestions. There were however a few exceptions, that we explain below.

Concerning the length of the manuscript, we have put the analytical methods as an Appendix 6 and we have tried to reduce a little the text. However we are convinced that we have to keep all the figures with the detailed mineralogical descriptions (which have been highly appreciated by all the reviewers), otherwise it will be very tidy for the reader to open each times the annex to find the relevant figure corresponding to the petrographic detail he would like to verify, and they are also important as we are using diagrams which are not only geochemical but mineralogical-chemical. For a good understanding of the trends defined in these diagrams the mineralogical observations are also essential. We think that such detailed mineralogical investigation combined with mineral-chemical diagrams is rare in the literature and thus should be preserved as such !

Quality of the microphotographs: you have had some difficulties to examine the microphotographs, and especially those from Figure 18 with the images of cathodoluminescence, in the pdf version of the manuscript. The resolution of the pictures is very bad in the pdf and explain some of their remarks! We hope that this problem will be alleviated when publishing.

The conditions for the quantitative analysis of major elements in accessory minerals in the analytical technique section has been precized.

Comment on lines 1008-1012. There is a concern about the status of the first alteration event recorded in the G1 suite. You are using a generalized concept of what is a greisen, whereas in our work we insist to only retain the most obvious definition of greisen as the result of a metasomatic process involving H+ and the destruction of feldspar, yielding a quartz-muscovite association, the equivalent in the Sn-W realm of the sericitic alteration in the copper porphyry systems. It happens that the first alteration event in G1 only involved protolithionite, the feldspars remaining unaffected: it therefore cannot be assigned to greisenisation. Being a kind of reequilibrating under submagmatic temperatures (muscovite + siderite ± chlorite assemblage), we think that this alteration is effectively more akin to propylitisation, although, admittedly, the process was not really isochemical, obviously involving some leaching of the ferromagnesian components.

Use of the word “matrix”. We have maintained the use of this word throughout, because “groundmass” is more specific, in our opinion, of volcanic rocks. In addition, the expression “matrix supported” is standard in breccia descriptions.

Use of the word “quartzification”. We have also maintained the use of this word, because (although admittedly of rare use in current literature), it seems to us the more accurate or the description of the addressed process.

Reviewer 3 Report

The paper represents very detailed data on petrography, mineralogy and geochemistry of the concealed Panasqueira granite intrusion and interpretation of its role as a metal source for world class Panasqueira Sn-W-Cu deposit. The results are of high importance for understanding of genesis of large rare-metal deposits of greisen formation. The majority of materials and discussion is based on mineralogy and petrography and thus the topic is suitable for publication in the ‘Minerals’ journal.

The authors build their approach to interpretation on the observation of major minerals in thin sections, mineral zonation and interrelations on the micron scale in SEM BSE images, and abundant chemical analyses of minerals and rocks. The data, represented in the paper, supports in general the major conclusion that metasomatic process is not related to granitic magma degassing (magmatic-hydrotherma transition), and occurred long after the solidification of the sampled part of the intrusive. The data also implies that the metasomatising fluids leach ore components important for Pnasqueira ores out of solidified intrusive.

The paper is well organized but the descriptions are too long on my mind, and I would suggest moderate shortening of the text. I would suggest to authors making double-check of Figures, as many of them require corrections (see remarks on figures in the PDF file).

Some criticism and technical comments are given below.

Page 3, Figure 1 legend. Better, to use ‘thermally metmorphosed rocks’, or ‘spotted schists’ as they will be called below, instead of ‘thermal metamorphism’.

Page 4, Figure 2. There is no caption for Figure 2C. Figure 2B: That is not a 3D model. The figure shows gravity anomaly, related to Panasqueira granite body. It can be transformed into 3D model, but this picture does not represent the model itself.

Page 5, Figure 3. Please check captions. It looks like the authors entangled the captions.

Page 5-6. The authors use G1 abbreviations before it first time appears in the text (beginning of the page 6). I would suggest moving the first paragraph of the Page 6 before the last one in the Page 5. The first paragraph of the Page 6 describes primary granitic units, while the previous is focused more on greisens and other minor intrusive things. In page 5 line 143 G1 granites are called porphyric, in the line 147 – they are ‘biotite granite’ and the beginning of the page 6 contain mentioning of G1 as ‘porphyric two-mica granite’. The description requires unification. Probably G1 unit is composed of different granites, e.g. biotite and two-mica granites. If so, it should be stated clearly.

Page 6. I would suggest placing the details of method to electronic supplementary materials. Lines 182-183: there are two different sets of EMPA conditions. It is necessary to show tasks for which these different conditions were applied. It is especially important as the totals of most of analyses are outside of common reliability range (98-101%) (see comments for Appendix A1-A3 below). Author state that ‘Quantitative analysis of major elements in accessory minerals was performed on the JEOL J7600F instrument, using the same standards as for EPMA analyses’. The use of probe currents as high as 12-150 nA for EDS analyses on the field emission SEM may lead to strong instability of minerals under the probe due to high current densities.

Page 8, Figure 5. It is not clear, where the major objects are located. Please show the important places by arrows.

Page 9, Figure 6. Some of photos are non-informative. Incorrect scales. Please, refer to comments in the PDF file.

Page 10, Figure 7 caption. What is the RMG? This abbreviation was previously used for Agremela rare-metal granite only. Panasqueira granite indeed can be classified as rare-metal granite. However, here please use Panasqueira granite for conformity with previous text.

Page 15, line 428-429. Below in the text biotite and protolithionite are intermixed. the authors need to decide here how they call this mineral. Both names are correct. Nevertheless, their intermixing lead to misunderstanding of what is this text about.

Page 21, Figure17 caption. Wolframoixiolite is discredited mineral species. Better to use W-rich ixiolite.

Page 23, Figure 18. Some of the pictures (A, C and E-I) look like they lost their images. Some scales are also lost.

Page 24, line 676. The Monier and Robert diagram is not widely used for representation of the Li-mica compositions. For example, zinnwaldite composition in this diagram is similar to any siderophyllite-eastonite series mineral. As the authors have estimated Li concentrations, they can easily plot them into Tischendorf et al. (2004) diagram. The necessary calculations are already done in the Appendix A1. I would suggest using Tischendorf et al. (2004) diagram in the paper.

Page 26, line 775. Nb end-member of the tantalite-columbite series is called columbite, not tantalite. Thus, the name colombotantalite is not correct for the described minerals. Better to call these minerals as minerals of the tantalite-columbite series.

Page 29, line 853-854. The parameter A is written here and in many other places as Al-(K-Na-2Ca). It is not correct. Parameter A=Al-(K+Na+2Ca) or Al-K-Na-2Ca (in gram-atoms or atoms).

Page 29, line 871. Probably, in the phrase ‘Th increases upwards from 10 to 5 ppm’ ‘decreases’ should be instead of ‘increases’.

Page 31, Figure 25B. In this diagram and other de La Roche diagrams authors use color-coded bold lines for representation of fractionation trends for the Panasqueira granite. Positions of these lines look reasonable for G1 and to some extent for G2, as they are close to compositions of the bulk rocks, which were not strongly affected by greisenization. Nevertheless, they are not well justified in the text for G3 and G4 units, which are strongly greisenized or converted into greisens. The authors need to explain how they estimated de La Roche parameters for G3 and G4 granites (see comments for Page 39).

Page 36, line 1008-1012. I agree with authors’ idea that the Panasqueira greisenising fluids were not derived from magmatic-hydrothermal transition of the Panasqueirs G1-G4 granite. However, I cannot agree that early replacement of protolithionite was a process different from the major greisenisation. The greisenisation affects not only feldspar (plagiocalse), but also biotite mica, as femic phases become unstable under acidic metasomatism (Fe and Mg are removed from granites in the course of greisenization). Thus, replacement of early protolithionite by Ms0 could also be a part of the greisenization process. It is not a classical propilitization, because this process stabilizes sodium plagioclase and epidote along with carbonate and zeolites (Na-Ca assemblage). It is important that the early (and the most later) stages of greisenization could also stabilize albite, which becomes unstable at the peak of greisenisation (Shcherba, 1970). This can explain multiple recrystallization of plagioclase, which was described in this paper.

Page 37, line 1046. The early stage might be overprinted by later stages of the same process. That could be a reason why the authors do not see the early protolithionite replacement.

Page 37, line 1055. This inversion can be qualitatively read from Figure 25A inset, but I could not catch from this drawing the basis for quantitative estimation of specific points in the diagram (see comment to Figure 25).

Page 39, line 1135. Fractionation is a magmatic process, implying derivation of melts after crystallization of some minerals. Here authors consider two different processes, divided in time: magmatic crystallization and greisenisation. Thus, the process cannot be simply called 'fractionation'.

Appendix A1-A3

In many cases totals of SEM EDS and EMPA analyses are more than 100% or less than 99%. Usually such data are excluded from interpretations. Assuming amounts of points in Figures, I can conclude that all the data were used in interpretations. I believe that the authors have to discuss the problem of high/low analytical totals and reliability of analyses.

Use of F or Si contents for estimation of Li contents in mica is a common approach, but frequently it gives odd values. Better to use charge balance for verification of Li by F/Si. For this, calculation of formula coefficients could be done to one alkali cation (K+Na+Rb+Cs).

Appendix A6

The system used by the authors is very complex. However, it really gives very useful regularities. As it is not widely used, I would focus authors to give more detailed explanations for vector calculation procedure. It is necessary to add formula for M3. The vector M3 in Figure A5 has strange scale. I was not able to derive formula for M*3 simply on the statement of authors that it is symmetrical to B*3.

As a summary, I believe that this paper can be published in the ‘Minerals’ journal after some shortening and minor revisions.

Author Response

We have carefully taken into account most of your suggestions, we warmly thank for your attentive reading, and the pertinent remarks and suggestions. There were however a few exceptions, that we explain below.

Concerning the length of the manuscript, we have put the analytical methods as an Appendix 6 and we have tried to reduce a little the text. However we are convinced that we have to keep all the figures with the detailed mineralogical descriptions (which have been highly appreciated by all the reviewers), otherwise it will be very tidy for the reader to open each times the annex to find the relevant figure corresponding to the petrographic detail he would like to verify, and they are also important as we are using diagrams which are not only geochemical but mineralogical-chemical. For a good understanding of the trends defined in these diagrams the mineralogical observations are also essential. We think that such detailed mineralogical investigation combined with mineral-chemical diagrams is rare in the literature and thus should be preserved as such !

Quality of the microphotographs: you have had some difficulties to examine the microphotographs, and especially those from Figure 18 with the images of cathodoluminescence, in the pdf version of the manuscript. The resolution of the pictures is very bad in the pdf and explain some of their remarks! We hope that this problem will be alleviated when publishing.

The conditions for the quantitative analysis of major elements in accessory minerals in the analytical technique section has been precized.

Comment on Fig 19. We do not agree with the suggestion of Reviewer3 for the Figure 19 “This diagram is not convenient for interpretation of the mica compositions. Better to use more recent diagram proposed by Tischendorf et al., 2002, which represents both important features of mica composition and their classification.” We consider that the diagram of Monier and Robert is the best for our purpose because it is based on a hydrothermal experimental data, and the substitution vectors between end member compositions can be easily deduced (as shown in the detailed figure below). The diagram of Tischendorf et al., 2002 is totally empirical, the micas species are defined by fields not by end-member compositions, and some of the mica species used in the diagram are not recognized by IMA.

What is more, the Monier diagram does not require any estimation of the Li content, being to the contrary a unique way to decipher the actual presence of Li in a given mica. In addition, we do not understand your concern with the siderophyllite-eastonite solid solution which concern the Mg content of the micas and which are not very different in the different mica generations.

Comment on Fig. 25B. You 3 wish to know how were estimated the parameters for the magmatic trend for the G3 and G4 granites. It is a good question, and we have improved the text for a better answer. Basically, we started from the final compositions and inverted the sequence of alteration processes. This is of course at the best semi-quantitative, so we also were guided by (i) the part of the trend quite well shown by G1-G2, already indicating that we are dealing with a RMG trend, and (ii) by the comparison with known RMG trends (such as Beauvoir RMG).

Comment on lines 1008-1012. There is a concern about the status of the first alteration event recorded in the G1 suite. you disagree with our statement of a “propylitic” style of this alteration, arguing that it is more akin greisenization. In our work we insist to only retain the most obvious definition of greisen as the result of a metasomatic process involving H+ and the destruction of feldspar, yielding a quartz-muscovite association, the equivalent in the Sn-W realm of the sericitic alteration in the copper porphyry systems. It happens that the first alteration event in G1 only involved protolithionite, the feldspars remaining unaffected: it therefore cannot be assigned to greisenisation. Being a kind of reequilibrating under submagmatic temperatures (muscovite + siderite ± chlorite assemblage), we think that this alteration is effectively more akin to propylitisation, although, admittedly, the process was not really isochemical, obviously involving some leaching of the ferromagnesian components.

Comment on line 1046. We think that, should protolithionite (or, better, zinwaldite) has been present in the pristine assemblage, we would have seen relicts. We think more likely that there was a magmatic Li-Fe muscovite in the pristine assemblage as observed in non-altered equivalent RMG granites.

Comment on Appendix 1. Concerning analyses with less than 99% or more than 100%: as these are not “whole rock” style of analyses, we made use of them as far as they yielded structural formulas (i) well equilibrated (for example K+Na between 0.9 and 1.1 in micas), and (ii) not basically differing from those obtained from “good” analyses. Others were rejected.

Concerning Li, we agree that the empirical formulas often yield odd estimates. But we do not agree with the suggestion of using charge balance, because (i) it is particularly sensitive to small errors, and (ii) it depends on assumptions on the Fe oxidation state, the latter problem being by far the more concern.

Comment on Appendix 6. In the Q-P, you are surprised by the position of muscovite on the plagioclase-orthoclase line being on the left of the orthoclase projection, thinking that this would involve the presence of some Ca in the muscovite structure : what is not, because, owing to the fact that there is less K2O in muscovite than in orthoclase, the calculation yields less K millications in muscovite than in orthoclase, and, consequently, the P parameter (K-Na-Ca) yields lower values for muscovite than for orthoclase, although there is  no Ca nor Na in none of them.  Same thing for position of the « G point ».

Concerning the (QBF)3 diagram, you ask for a specific formula for the M*3 parameter. But, in fact, M*3 is not a new parameter: it is only a way to represent B*3 when the latter appears to be negative. The interest of using M*3 resides in the fact that pure muscovite plot onto the 100 M*3 pole.

Round 2

Reviewer 1 Report

The revised manuscript is now distinctly improved which is also true for the style of the language. Errors have been erased and the manuscript is distinctly shorter and clearer now. I can recommend the new manuscript for publication now.